# Blood and lymphatic systems are segregated by the FLCN tumor suppressor

Ikue Tai-Nagara[1], Yukiko Hasumi[2,3], Dai Kusumoto[4], Hisashi Hasumi[3,5], Keisuke Okabe[1,6], Tomofumi Ando[1,7], Fumio Matsuzaki[8], Fumiko Itoh[9], Hideyuki Saya [10], Chang Liu[11], Wenling Li[11], Yoh-suke Mukouyama[11], W. Marston Linehan [3], Xinyi Liu[12], Masanori Hirashima[12], Yutaka Suzuki[13], Shintaro Funasaki[14], Yorifumi Satou [15], Mitsuko Furuya[16], Masaya Baba [3,14,17✉] & Yoshiaki Kubota [1,17✉]

Blood and lymphatic vessels structurally bear a strong resemblance but never share a lumen, thus maintaining their distinct functions. Although lymphatic vessels initially arise from embryonic veins, the molecular mechanism that maintains separation of these two systems has not been elucidated. Here, we show that genetic deficiency of Folliculin, a tumor suppressor, leads to misconnection of blood and lymphatic vessels in mice and humans. Absence of Folliculin results in the appearance of lymphatic-biased venous endothelial cells caused by ectopic expression of Prox1, a master transcription factor for lymphatic specification. Mechanistically, this phenotype is ascribed to nuclear translocation of the basic helix-loop-helix transcription factor Transcription Factor E3 (TFE3), binding to a regulatory element of Prox1, thereby enhancing its venous expression. Overall, these data demonstrate that Folliculin acts as a gatekeeper that maintains separation of blood and lymphatic vessels by limiting the plasticity of committed endothelial cells.

[1] Department of Anatomy, Keio University School of Medicine, 35 Shinanomachi, Shinjuku-ku, Tokyo 160—8582, Japan. [2] Department of Ophthalmology, Yokohama City University Graduate School of Medicine, 3-9 Fukuura, Kanazawa-ku, Yokohama, Kanagawa 236-0004, Japan. [3] Urologic Oncology Branch, Center for Cancer Research, National Cancer Institute, National Institutes of Health, Bethesda, MD 20892, USA. [4] Department of Cardiology, Keio University School of Medicine, 35 Shinanomachi, Shinjuku-ku, Tokyo 160—8582, Japan. [5] Department of Urology, Yokohama City University Graduate School of Medicine, 3-9 Fukuura, Kanazawa-ku, Yokohama, Kanagawa 236-0004, Japan. [6] Department of Plastic Surgery, Keio University School of Medicine, 35 Shinanomachi, Shinjuku-ku, Tokyo 160-8582, Japan. [7] Department of Surgery, Keio University School of Medicine, 35 Shinanomachi, Shinjuku-ku, Tokyo 160-8582, Japan. [8] Laboratory for Cell Asymmetry, RIKEN Center for Biosystems Dynamics Research, 2-2-3 Minatojima-Minamimachi, Chuou-ku, Kobe 650-0047, Japan. [9] Laboratory of Cardiovascular Medicine, Tokyo University of Pharmacy and Life Sciences, Horinouchi, Hachioji, Tokyo 1432-1, Japan. [10] Division of Gene Regulation, Institute for Advanced Medical Research, Keio University School of Medicine, 35 Shinanomachi, Shinjuku-ku, Tokyo 160-8582, Japan. [11] Laboratory of Stem Cell and Neuro-Vascular Biology, Cell and Developmental Biology Center, National Heart, Lung, and Blood Institute, National Institutes of Health, Bethesda, MD 20892, USA. [12] Division of Pharmacology, Niigata University Graduate School of Medical and Dental Sciences, 1—757 Asahimachi-dori, Chuo-ku, Niigata 951-8510, Japan. [13] Department of Computational Biology and Medical Sciences, Graduate School of Frontier Sciences, The University of Tokyo, Kashiwa, Chiba 277-0882, Japan. [14] Laboratory of Cancer Metabolism, International Research Center for Medical Sciences, Kumamoto University, Kumamoto 860-0811, Japan. [15] Division of Genomics and Transcriptomics Joint Research Center for Human Retrovirus Infection, Kumamoto and Kagoshima Universities, Kumamoto 860-0811, Japan. [16] Department of Molecular Pathology, Yokohama City University Graduate School of Medicine, 3-9 Fukuura, Kanazawa-ku, Yokohama, Kanagawa 236-0004, Japan. [17] These authors contributed equally: Masaya Baba, Yoshiaki Kubota. ✉email: babam@kumamoto-u.ac.jp; ykubo33@a3.keio.jp

The two major circulatory systems of blood and lymphatic vasculature are discretely distributed throughout the body. While the former provides tissues with oxygen and nutrients, the latter drains the interstitial fluid from the tissue spaces to return it to the bloodstream[1], transports immune cells, absorbs dietary lipids[2], and clears waste from the brain[3]. The structures of the two systems, the blood vessel (BV) and the lymphatic vessel (LV), are histologically similar but anatomically do not share the same lumen, except at the lymphovenous valve located near the venous angle, the final junction of collecting lymph ducts, and subclavian veins[4,5]. LVs arise mostly from preexisting veins[6,7], although populations of lymphatic progenitors of non−venous origin have been identified[8,9]. The expression of Prox1, a master transcription factor controlling lymphatic specification, determines both the initiation and maintenance of the identity as a lymphatic endothelial cell (LEC), contributing to the separation of the blood and lymphatic systems[10,11]. Platelets activated by committed LECs also contribute to this separation[5,12,13]. However, the molecular mechanisms that ultimately maintain separation of these two circulatory systems have not been fully identified.

Folliculin (*FLCN*) has long been studied as a tumor suppressor gene responsible for the human autosomal dominant genetic disorder Birt-Hogg-Dubé (BHD) syndrome[14]. Affected individuals are at risk of developing of kidney cancers, lung cysts, and cutaneous fibrofolliculomas[15,16]. FLCN encoded by the *FLCN* gene, as a complex with its binding proteins FNIP1/2, acts as a GTPase activating protein (GAP) for Rag GTPases, thereby transducing amino acid levels to mTOR complex 1 (mTORC1)[17−19] and secluding the basic helix-loop-helix (bHLH) transcription factor and member of the MiTF family, Transcription Factor E3 (TFE3), in the cytoplasm[20,21]. Recently, FLCN has been revealed as the central player in the control of pluripotency of embryonic stem (ES) cells[21,22]. In this study, based on a unique phenotype of *Flcn*−deficient mice, we have identified the fundamental mechanism maintaining the separation of the blood and lymphatic vascular systems through limiting the plasticity of committed endothelial cells.

## Results

**Flcn deficiency causes blood-filled and dilated lymphatics.** *Flcn* deficiency in mice causes visceral endoderm defects and early embryonic lethality at E5.5–E6.5[23]. To achieve endothelial-specific knockout of *Flcn*, we first utilized *Tie2–Cre*, known to recombine genes in all endothelial and hematopoietic cells[24]. *Tie2–Cre*-specific *Flcn*-knockout mice (*Tie2–Cre+Flcnflox/flox*, hereafter referred to as *FlcnΔEHC*) died before E15.5 (Fig. 1a), and showed hemorrhagic appearance. Histologically, *FlcnΔEHC* embryos showed apparently dilated and blood-filled LVs at E14.5 (Fig. 1b–f). In agreement with the region-specific differences in *Tie2–Cre* expression[8], LV dilation was apparent in thoracic but not lumbar skins of *FlcnΔEHC* embryos (Fig. 1g–l). To test endothelial *Flcn* function more specifically, we deleted *Flcn* postnatally using *Cdh5-BAC-CreERT2* mice (*Cdh5-BAC-CreERT2+Flcnflox/flox*, hereafter referred to as *FlcniΔEC*). *FlcniΔEC* pups died with reddish colored skin (Fig. 1m, n) around 7 days after 4-hydroxytamoxifen treatment (at postnatal (P) day 9). *FlcniΔEC* pups showed blood-filled and dilated LVs (Fig. 1o, x). We analyzed *FlcniΔEC* embryos at E15.5 (Supplementary Fig. 1a–e), and found they recapitulated the defects of *Tie2–Cre+Flcnfl/fl* embryos. However, we did not detect such defects in adult *FlcniΔEC* mice treated with tamoxifen after weaning (Supplementary Fig. 1f-l). Taken together, *Flcn* deficiency caused blood-filled and dilated lymphatics during embryogenesis and postnatal development.

**Flcn deficiency leads to ectopic Prox1 expression in veins.** Next, we established a technique to visualize whole-mount tail skins (Fig. 2a–e; Supplementary Fig. 2a–k). This technique allowed us to easily visualize two streams of arterial-venous-lymphatic bundles on both sides of the tail, and *Cdh5-BAC-CreERT2* excised *Flcnflox* alleles in both blood and lymphatic endothelial cells (Supplementary Fig. 2l). With this technique, the lymphatic dilation in *FlcniΔEC* mice was apparent (Fig. 2b, c). Notably, Prox1 was ectopically expressed in VECs of *FlcniΔEC* mice, some even with equal intensity to that seen in their LECs (Fig. 2d–j). Prox1 expression was also observed in retinal veins and the vena cava in *FlcniΔEC* pups and cervical veins of *FlcnΔEHC* embryos (Fig. 2k–p), suggesting that this phenomenon is widespread. When we deleted *Flcn* in adult endothelial cells, we observed similar ectopic expression of Prox1 in VECs (Supplementary Fig. 1m, n). Considering that we did not detect lymphatic hyperplasia and blood filling in adult *FlcniΔEC* mice, some compensation mechanism might have occurred after weaning and masked the phenotypes seen in their neonatal period. Consistent with the knowledge that transgenic Prox1 overexpression induces an LV phenotype in VECs[25], mature LEC markers like Pdpn and Vegfr3[26] were slightly but significantly upregulated in Prox1+ VECs of *FlcniΔEC* mice (Supplementary Fig. 2m–o). In high magnification views, we observed abnormal sprouting from LVs toward veins as well as from veins toward LVs in *FlcniΔEC* mice (Supplementary Fig. 2p–u). We also detected occasional bridging of endothelial filopodia between LVs and veins in *FlcniΔEC* mice (Supplementary Fig. 2u), suggesting they somehow attracted each other. The EdU incorporation assay showed increased LEC but not VEC proliferation in *FlcniΔEC* mice (Fig. 2q–v). This hyper proliferation of LECs is likely responsible for lymphatic enlargement, and is caused by increased Vegfr3 expression (Supplementary Fig. 2m–o) enhancing ligand responsiveness.

**Flcn deficiency gives rise to lymphatic-biased venous endothelial cells.** To elucidate more precisely the cellular status, we employed single-cell RNA sequencing (scRNA-seq) on CD31+ cells isolated from the mesentery (Supplementary Fig. 3a, b). The Uniform Manifold Approximation and Projection (UMAP) of cells revealed interconnected distribution of gene expression profiles suggesting continuities and discontinuities between each cell type (Fig. 3a). Subsequent clustering analysis identified non-endothelial and six endothelial clusters (Fig. 3b, c) that could be assigned as capillary, venous, or arterial endothelial cells (one cluster each) or LECs (three adjacent clusters) based on the expression of known endothelial subtype markers (Fig. 3c, d; Supplementary Fig. 3c). Most of the CD31+ non-endothelial cells were hematopoietic cells as they expressed CD45 (Supplementary Fig. 3c–e). Notably, cells from *FlcniΔEC* but not control mice showed a unique cell population within VECs, which expressed Prox1 (Fig. 3e–i). Comparison of transcriptional profiles revealed that this Prox1+ VECs most resembled VECs and showed a transitional state between a venous and lymphatic endothelial fate in terms of those specific markers (Fig. 3j, k; Supplementary Fig. 4a). Considering the elevated expression of mature lymphatic markers in veins of *FlcniΔEC* mice (Supplementary Fig. 2m–o) we named this population as LEC-biased VECs. Among those listed genes, we confirmed the upregulation of Cldn11 and downregulation of vWF in LEC-biased VECs, namely, VECs found in *FlcniΔEC* mice, by immunohistochemistry (Supplementary Fig. 4b–h). Extraction of genes characterizing the cluster of Prox1+ VECs did not highlight any gene specifically expressed in this cell population; rather, most were strongly expressed in VECs and LECs (Supplementary Fig. 4i, j; Supplementary Data 1 and 2).

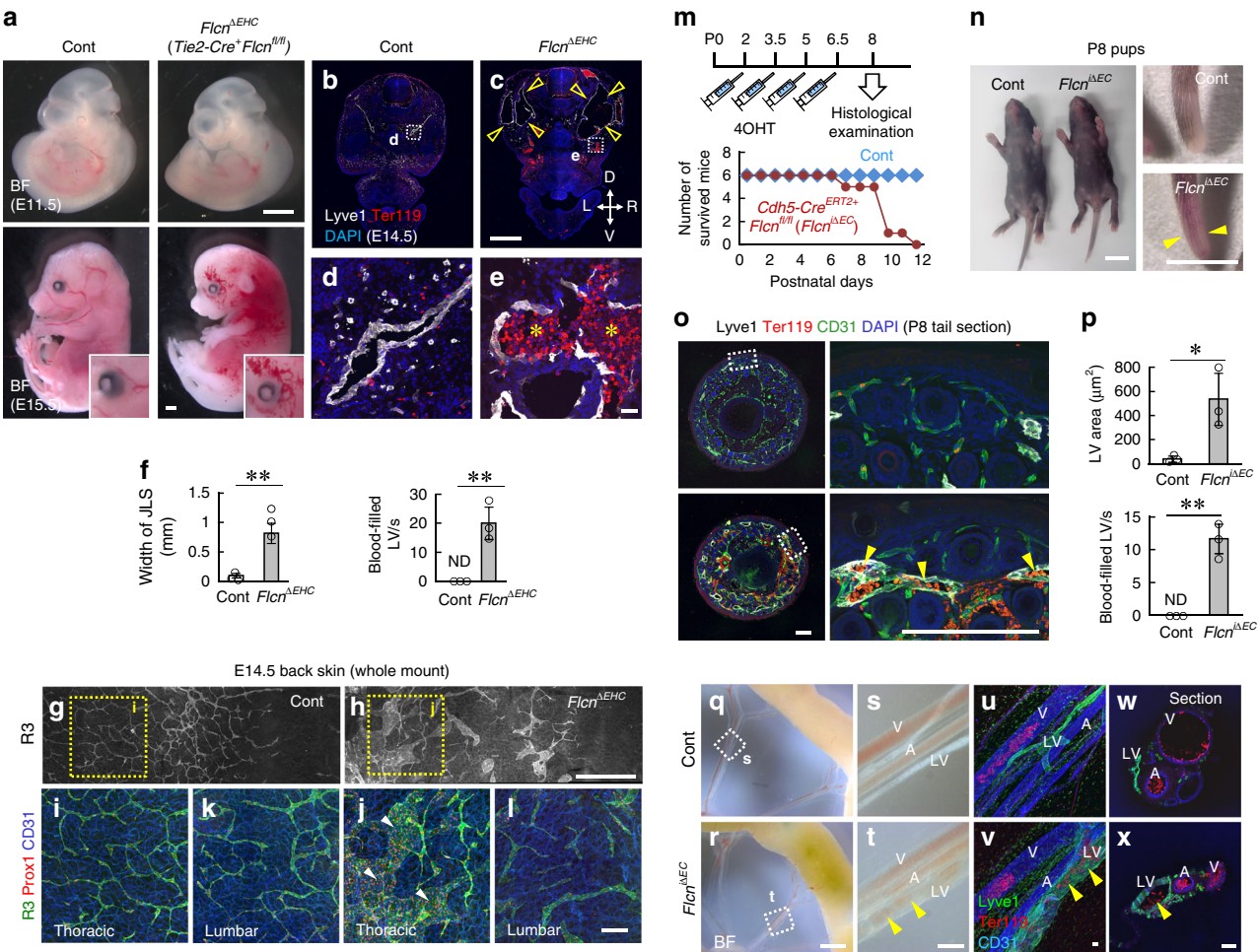

**Fig. 1 Flcn deficiency causes blood-filled and dilated lymphatics. a** Images of $Flcn^{\Delta EHC}$ mice at embryonic day (E)11.5 or E15.5. **b–l** Transverse sections (**b–e**) or whole-mount skin samples (**g–l**) of E14.5 embryos and quantification (n = 3). $Flcn^{\Delta EHC}$ embryos show dilated (open and closed arrowheads) and blood-filled (asterisks) lymphatic vessels. **m** Protocol for 4-hydroxytamoxifen (4OHT) injection in neonates and survival curve of control and $Flcn^{i\Delta EC}$ mice (each n = 6). P postnatal. **n–p**)Control and $Flcn^{i\Delta EC}$ pups and tail sections at P8, and quantification after staining with the indicated antibodies (n = 3). $Flcn^{i\Delta EC}$ mice show reddish colored skin (arrowheads in **n**) as well as dilated and blood-filled LVs (arrowheads in **o**). **q–x** Bright field views and whole-mount or section samples of mesentery at P8. Erythrocytes (closed arrowheads) are detected not only in arteries (A) and veins (V) but also in LVs of $Flcn^{i\Delta EC}$ mice. Scale bars: 5 mm (**o**); 1 mm (**a–c, g, h, q, r**); 200 µm (**i–l, o, s, t**); 50 µm (**d, e, u–x**). **P < 0.01; *P < 0.05; ND not detected. Data presented are the mean ± SD. The comparisons between the averages of the two groups were evaluated using the two-sided Student's t-test. Source data are provided as a Source Data file.

As confirmation, immunostaining revealed Prox1 expression in CoupTFII⁺Emcn⁺ VECs in $Flcn^{i\Delta EC}$ mice (Fig. 3l).

**Ectopic Prox1 in veins accounts for the BV-LV misconnection in Flcn^{iΔEC} mice.** In agreement with the lymphatic blood filling in $Flcn^{i\Delta EC}$ mice, intracardiac injection of lectin revealed its influx into LVs, suggesting that somewhere BVs and LVs were mixed (Fig. 4a). In section specimens, LVs and veins shared a lumen perfused with lectin in $Flcn^{i\Delta EC}$ mice (Fig. 4b–h). Transgenic overexpression of Prox1 in embryonic endothelial cells also resulted in similar blood-filled lymphatics (Fig. 4i–o; Supplementary Fig. 5). $Flcn^{i\Delta EC}$ mice showed normal zipper-like junctions in LVs[2,26], no abnormal recruitment of smooth muscle cells around LVs (Supplementary Fig. 6a–h). At P6, venous Prox1 expression and blood-filled lymphatics were already evident in $Flcn^{i\Delta EC}$ mice without subcutaneous hemorrhage (Supplementary Fig. 6i–l). These data suggest that erythrocytes, which entered LVs, eventually extravasated from the inter-cellular gap between

LECs. Next, we examined the lymphovenous valve (LVV) located at the venous angle as the cause of lymphatic blood filling. In immunohistochemistry of LVVs, we could not detect apparent abnormalities of the LVV structure including the two-layer structure of VECs and LECs in $Flcn^{i\Delta EC}$ mice at P8 (Fig. 4p; Supplementary Fig. 6m–r). Under a stereo microscope, no apparent blood filling was observed in the thoracic ducts (TDs) of either control or $Flcn^{i\Delta EC}$ mice (Supplementary Fig. 6s–v), suggesting impaired function of the LVV is less likely to be the cause of lymphatic blood filling in $Flcn^{i\Delta EC}$ pups. Importantly, Prox1 deletion in endothelial cells normalized the phenotypes of $Flcn^{i\Delta EC}$ mice (Fig. 4q–t), suggesting their aberrant Prox1 expression was causative. Interestingly, mice lacking Prox1 in endothelial cells showed absence of lymphatic valves, although they did not have apparent impairment in overall structures of lymphatic vascular networks in accordance with the strong expression of Prox1 in lymphatic valves at this stage. The LEC-specific Flcn deletion ($Flcn^{i\Delta LEC}$) led only to lymphatic hyperplasia but not blood-filled LVs and lethality (Fig. 4u–z),

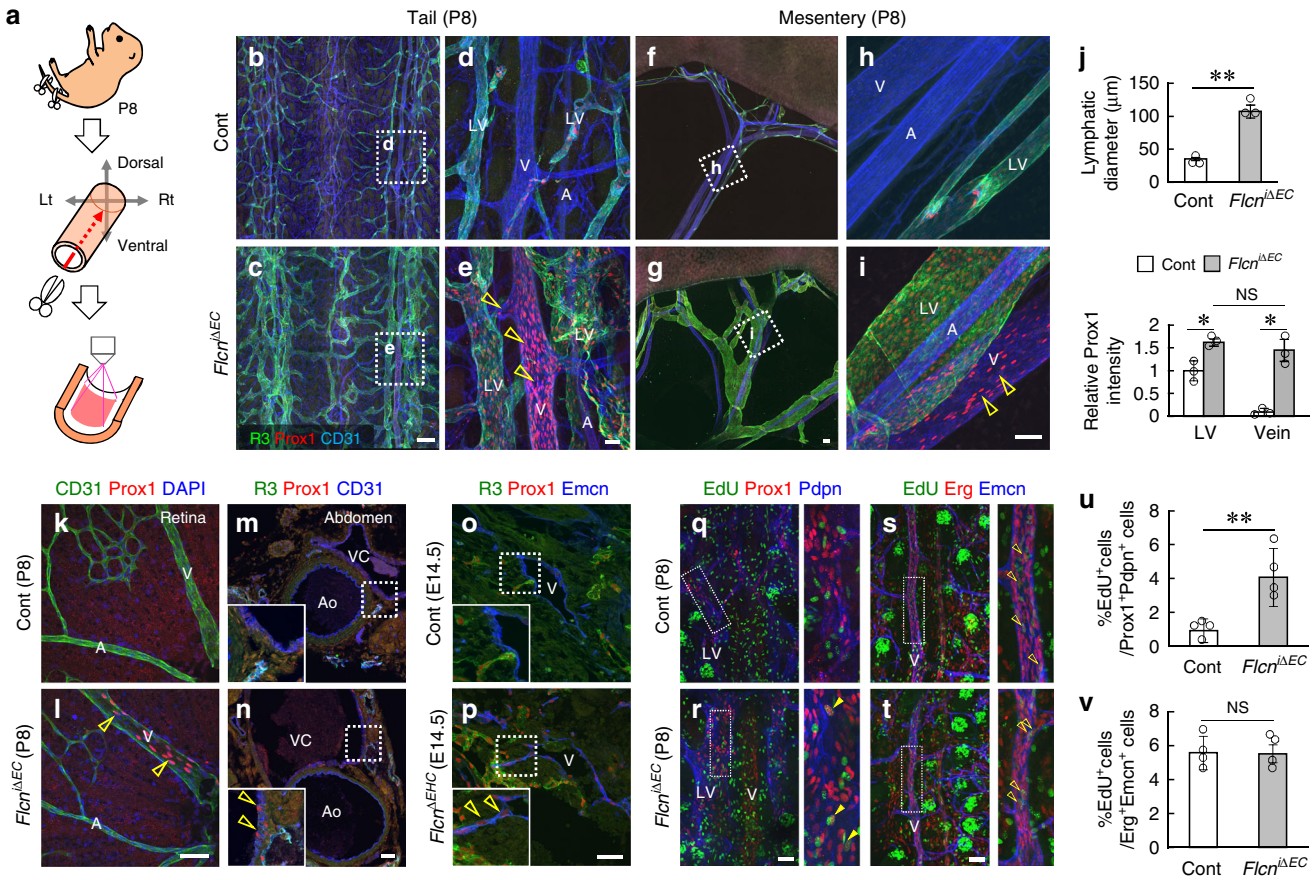

**Fig. 2 Flcn deficiency leads to ectopic Prox1 expression in veins. a** Schematic diagram depicting the technique used to prepare whole-mount tail samples.
**b–j** Tail or mesenteric whole-mount samples at P8, and quantification in tails ($n = 3$). $Flcn^{i\Delta EC}$ mice show ectopic Prox1 expression (open arrowheads) in veins (V) and its branches. LV, lymphatic vessels; A, arteries. **k–p** Retinal whole-mounts or abdominal sections at P8 and embryonic sections at E14.5. $Flcn^{i\Delta EC}$ and $Flcn^{\Delta EHC}$ mice show ectopic Prox1 expression (open arrowheads) in veins (V) but not arteries (A). VC vena cava, Ao abdominal aorta. **q–v** Immunohistochemical analysis of tail whole-mounts at P8 and quantification ($n = 4$). LVs show increased endothelial proliferation (closed arrowhead) in $Flcn^{i\Delta EC}$ mice. Open arrowheads indicate proliferation in venous endothelial cells. Scale bars: 200 μm (**b, c, f, g**); 50 μm (**d, e, h, i, k–t**). **$P < 0.01$; *$P < 0.05$; NS not significant, ND not detected. Data are presented as the mean ± SD. The comparisons between the averages of the two groups were evaluated using the two-sided Student's $t$-test. Source data are provided as a Source Data file.

suggesting that the latter phenotype in $Flcn^{i\Delta EC}$ mice might be attributed to ectopic Prox1 expression in VECs.

**TFE3 binds the regulatory element of PROX1 directly.** To determine the mechanism for Prox1 upregulation in $Flcn^{i\Delta EC}$ mice we focused on Tfe3, which intracellular localization and degradation is well known to be regulated by Flcn[21,22]. In the wild-type tail skin, Tfe3 immunoreactivity was abundant in mesenchymal cells, but barely visible in BVs or LVs (Fig. 5a, b). However, the VECs and LECs in $Flcn^{i\Delta EC}$ mice showed strong expression of Tfe3, which was localized in endothelial nuclei (Fig. 5c–l). The specificity of this staining was confirmed using *Tfe3* deficient mice (Supplementary Fig. 7a–d). Quantitative PCR analysis indicated that the mRNA expression of *Tfe3* in isolated BECs was not different between control and $Flcn^{i\Delta EC}$ mice (Fig. 5m), suggesting nontranscriptional regulation like nuclear translocation and lysosomal degradation[21,22] is responsible for increased Tfe3 immunoreactivity. In control veins, Tfe3 proteins were colocalized with RagC, suggesting that Flcn secludes Tfe3 in the cytoplasm by binding the GDP form of RagC/D (Fig. 5n–u; Supplementary Fig. 7e–n), as has previously been found in embryonic stem cells[17,21]. Analysis of scRNA-seq data showed expressions of *Flcn* and *Tfe3* were significantly higher in VECs than arterial endothelial cells (Supplementary Fig. 3f). This

expression tendency might be, at least in part, the determinants for the VEC-inclined phenotype in $Flcn^{i\Delta EC}$ mice. Similarly, LECs had more abundant expression of *Flcn* than arterial endothelial cells, but the expression of Tfe3 was comparable between them (Supplementary Fig. 3f), suggesting nontranscriptional regulation of Tfe3 like nuclear translocation and lysosomal degradation. We performed chromatin immunoprecipitation sequencing (ChIP-seq) analysis using HK-2 cells. We found that TFE3 bound to the 150 bp genomic sequence just upstream of the transcription start site of PROX1 (hereafter referred to as PROX1−0.15 kb; Fig. 5v), which is distinct from the previously identified binding regions for Sox18, Gata2, FoxC2, and Nfatc1[27,28]. PROX1-0.15 kb has two E-box (enhancer box) sequences, suggesting a function as a regulatory element driven by a bHLH transcription factor (Fig. 5w). Indeed, doxycycline-induced TFE3 overexpression in HEK293 cells increased the expression of PROX1 (Fig. 5x), although the late induction of PROX1 may suggest the contribution of some indirect effect. The luciferase reporter assay confirmed the significance of E-box sequences located in PROX1-0.15 kb during TFE3-driven PROX1 transcription (Fig. 5y).

**Flcn suppresses Prox1 expression through Tfe3.** We used human umbilical vein endothelial cells (HUVECs) to analyze the functional role of TFE3 in FLCN-regulated PROX1 expression in

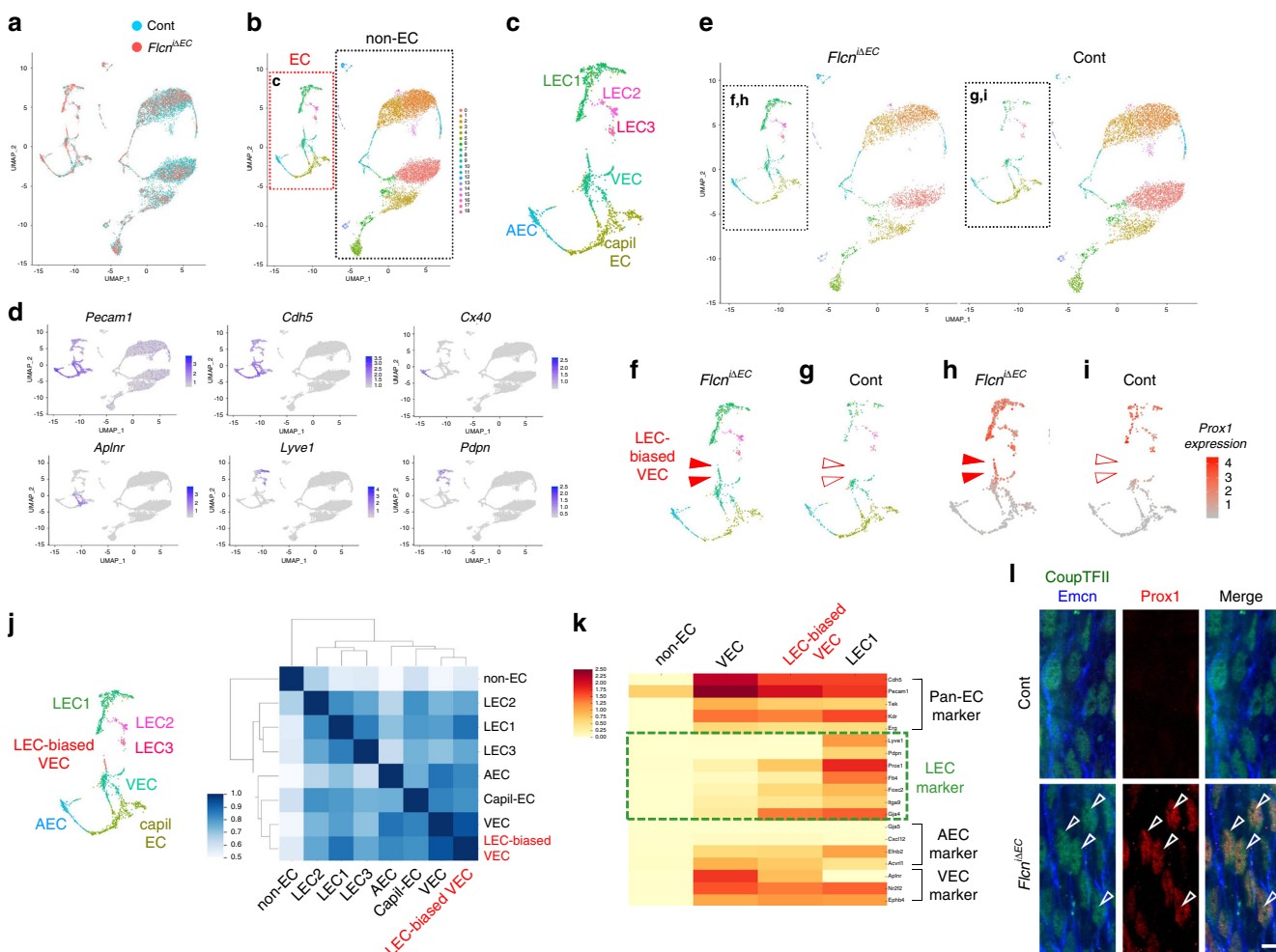

**Fig. 3 Flcn deficiency gives rise to lymphatic-biased venous endothelial cells.** Single-cell RNA-seq of CD31+ cells from the mesentery at P8. Pooled CD31+ cells isolated from three mouse mesenteries per genotype. **a–c** UMAP plots showing 6814 control and 4459 *Flcn^iΔEC* cells corresponding to non-endothelial (non-EC), arterial endothelial (AEC), capillary endothelial (capilEC), venous endothelial (VEC), and lymphatic endothelial (LEC1–3) cells. Two datasets (control and *Flcn^iΔEC* cells) are integrated by identifying anchor genes. **d** UMAP plot combining control and *Flcn^iΔEC* cells showing expression of marker genes with enriched expression for each endothelial cluster. **e–i** UMAP plots showing cells from control and *Flcn^iΔEC* mice. The cluster of lymphatic-biased venous endothelial cells (LEC-biased VECs) is detected in *Flcn^iΔEC* (closed arrowheads) but not control (open arrowheads) mice. Panels **h, i** show the Prox1 expression levels as indicated by the color key. **j** UMAP plot indicating LEC-biased VECs and the Pearson correlation matrix of the average expression profiles of all subpopulations based on all differentially expressed genes. **k** Heatmap showing marker genes for LECs and other populations. Rows represent genes and columns represent cells. **l** Tail whole-mounts at P8. *Flcn^iΔEC* mice show ectopic expression of Prox1 in VECs (arrowheads). Scale bar: 5 μm.

endothelial cells. ChIP analysis of HUVECs confirmed that endogenous TFE3 binds PROX1-0.15 kb (Fig. 6a, b). Next, we performed the same ChIP assay on HUVECs treated with small interfering (si)RNAs targeting *TFE3*, confirming that this binding occurred specifically in endogenous TFE3 (Fig. 6c, d). To evaluate the impact of the FLCN-TFE3 axis on the expression of PROX1, we knocked down *FLCN* with or without *TFE3* in HUVECs. We found si-*FLCN* significantly increased *PROX1* transcription, and its effect was normalized by si-*TFE3* (Fig. 6e, f, g). Supporting this observation, with si-*FLCN* treatment, TFE3+ cells largely correspond to PROX1+ cells (Fig. 6h–m). *Tfe3* deletion in vivo completely normalized blood-filled LVs, and partially normalized the lymphatic hyperplasia of *Flcn^iΔEC* mice (Fig. 6n–w). Flcn is reported to negatively regulate PGC1α and mitochondrial biogenesis in several organs, and biochemically functions in association with mTORC1[14,17,21,29]. Moreover, Prox1 expression is suppressed by rapamycin in a lymphangiectasia model[30],

suggesting some interaction among Prox1, mTor, and Flcn. We, therefore, created mice with deleted *Flcn* and deleted *Pgc1α* or *Mtor* in endothelial cells (*Flcn;Pgc1α^iΔEC* and *Flcn;Mtor^iΔEC*), but the phenotype of these deficient mice was indistinguishable from that of *Flcn^iΔEC* mice (Supplementary Fig. 8).

**FLCN haploinsufficiency causes lymphatic blood filling in humans.** Next, samples from three BHD patients with pathogenic germline variants of *FLCN* were examined histologically. Lung specimens showed significantly more LVs than normal lungs or non-BHD (nonspecific) bullae (Fig. 7a–d). In BHD lung specimens, some D2-40+PROX1+CD31+LVs were filled with blood, and ectopic PROX1 expression in D2-40-VECs was detected (Fig. 7e–g). In addition, total lung lysates from BHD lungs showed increased expression of TFE3 and PROX1 (Fig. 7h). Fluorescent immunohistochemical staining revealed that TFE3

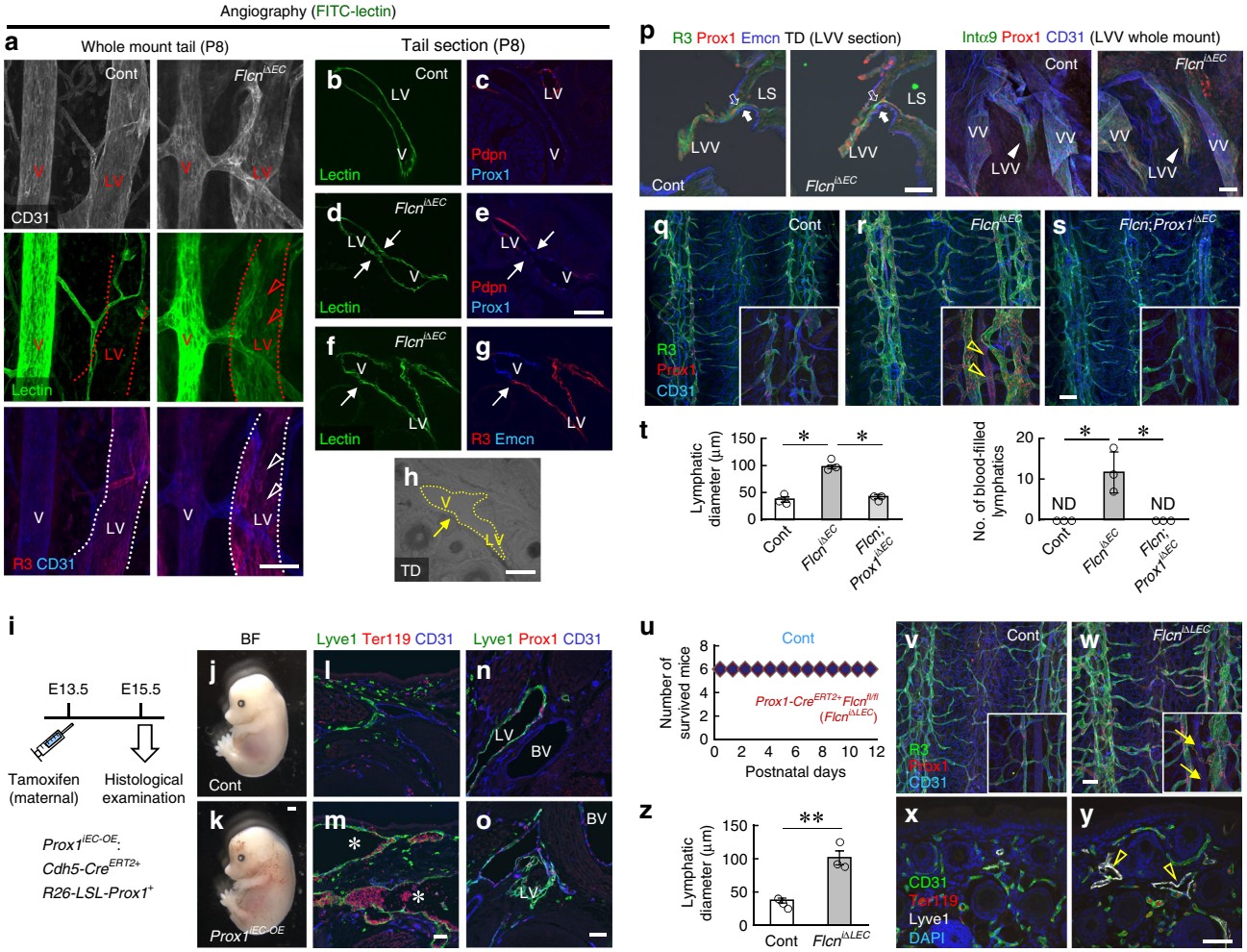

**Fig. 4 Ectopic Prox1 in veins accounts for the BV-LV misconnection in Flcn$^{i\Delta EC}$ mice. a–h** Whole-mount (**a**) or section (**b–h**) tail images taken after intracardiac lectin injection. *Flcn$^{i\Delta EC}$* mice show lectin perfusion into lymphatic vessels (LV) as well as veins (V) (arrowheads), and connection (arrows) between LVs and Prox1$^+$ veins. **i** Protocol for tamoxifen injection into pregnant females. **j, k** Images of control and *Prox1$^{iEC-OE}$* mice at embryonic day E15.5. **l–o** Immunohistochemical analysis of coronal sections of E15.5 embryos. *Prox1$^{iEC-OE}$* embryos show enlarged and blood-filled lymphatic vessels (asterisks), Prox1 is expressed in blood vessels (BV) as well as lymphatic vessels (LV) in *Prox1$^{iEC-OE}$* embryos. **p** Section or whole-mount immunohistochemistry of left venous angle at P8. No apparent abnormality is detected in lymphovenous valves (LVV; closed arrowheads) including the two-layer structure of venous (open arrows) and lymphatic endothelial cells (closed arrows) in *Flcn$^{i\Delta EC}$* mice. LS lymph sacs, VV venous valves. **q–t** Tail whole-mounts at P8 and quantification (n = 3). Ectopic Prox1 expression (open arrowheads) in veins of *Flcn$^{i\Delta EC}$* mice is attenuated by Prox1 deletion. **u** Survival curve of control and *Flcn$^{i\Delta LEC}$* mice (each n = 6). **v–z** Immunohistochemical analysis of tail whole-mounts (**v, w**) and sections (**x, y**) at P8, and quantification (n = 3). *Flcn$^{i\Delta LEC}$* mice lack ectopic Prox1 expression in veins (arrows) and erythrocytes in LVs (open arrowheads). Scale bars: 200 μm (**q–s, v, w**); 50 μm (**a–h, l–p, x, y**). The comparisons between the averages of the two groups were evaluated using the two-sided Student's *t*-test. **P < 0.01; *P < 0.05; ND, not detected. Data are presented as the mean ± SD. Source data are provided as a Source Data file.

was indeed detectable in nuclei of PROX1$^+$D2-40$^-$ VECs from BHD patients (Fig. 7i–n).

## Discussion

In this study, we found that strict limitation of plasticity between LECs and VECs governed by the FLCN-TFE3 pathway ensures the segregation of the lymphatic and blood circulatory systems. FLCN, as a complex with FNIP1/2, acts as a GAP for RagC/D and secludes TFE3 in the cytoplasm[21]. In our study of deficient mice, without *Flcn*, Tfe3 was translocated into the nucleus and ectopically upregulated Prox1 by binding to its promoter through the E-box in VECs (Fig. 7o).

Genetic regulators of Prox1 have been intensively studied. Sox18 directly activates Prox1 transcription by binding to its proximal promoter 4.5 kb downstream from the transcription

start site (Prox1 + 4.5 kb)[27]. In addition, a putative enhancer element of Prox1 (−11 kb upstream enhancer element) is bound by Gata2, Foxc2, and Nfatc1[28], genes identified to be important for lymphatic development[31,32]. Considering that *Tfe3*-knockout mice do not show lymphatic defects, Tfe3 binding to the Prox1− 0.15 kb might be dispensable for normal development, but may have an important role in pathological settings, in particular when Tfe3 is abnormally activated.

Mature LEC markers like Pdpn and Vegfr3 were slightly upregulated in LEC-biased VECs in *Flcn$^{i\Delta EC}$* mice, but were far weaker than in their LECs, suggesting Prox1 expression is not sufficient to fully convert VECs to LECs. Such an ambiguous endothelial population exists in the interface of BV-LV junctions. In the budding process of initial lymphatics, bipotential Prox1$^+$ precursor cells in veins asymmetrically divide into two daughters; one becomes Prox1$^{high}$ LECs, which start to express mature LEC

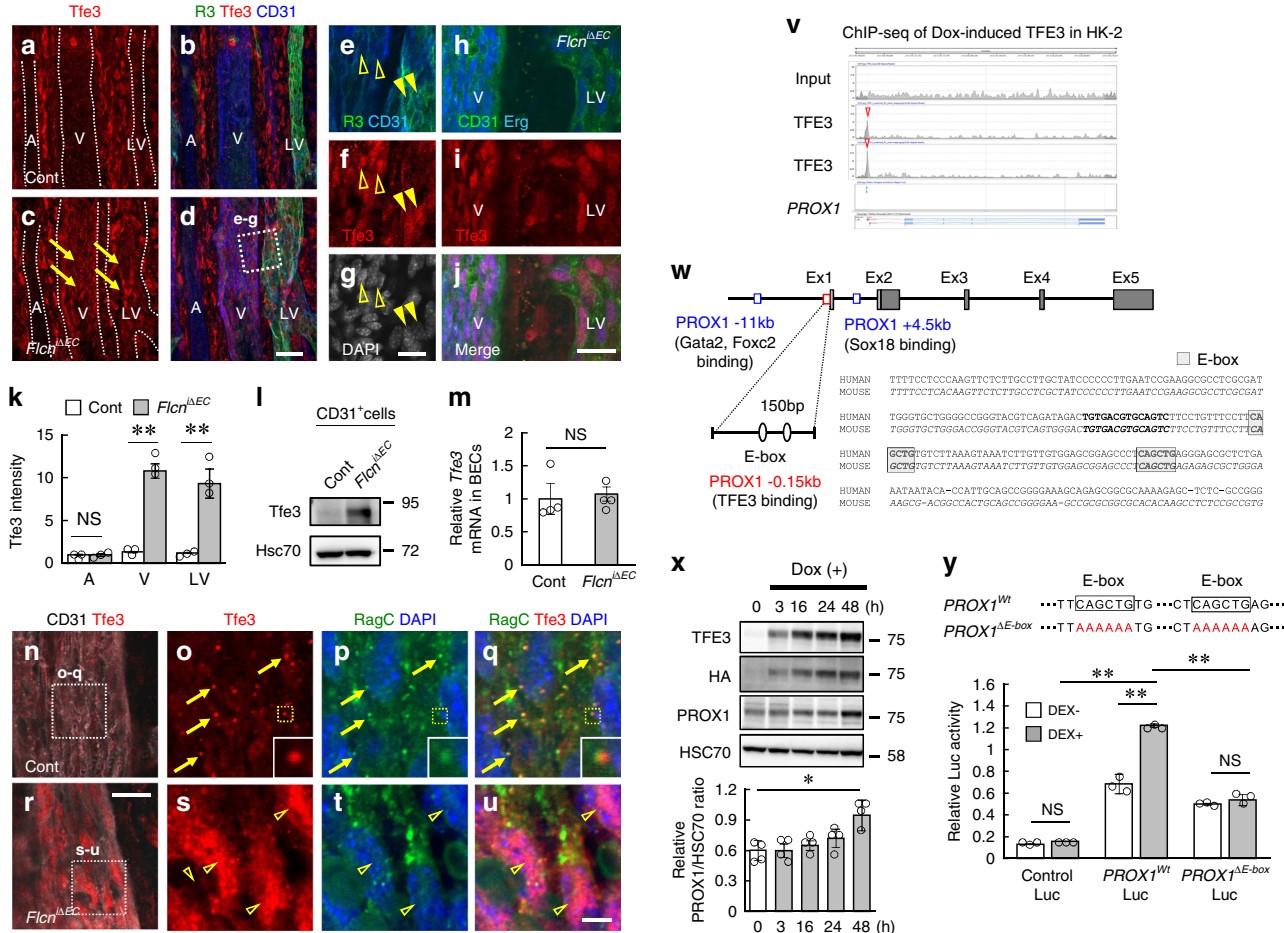

**Fig. 5 TFE3 binds the regulatory element of PROX1 directly. a–k** Tail whole-mounts at P8 and quantification ($n = 3$). Endothelial cells in veins (V) and lymphatic vessels (LV) but not arteries (A) of $Flcn^{i\Delta EC}$ mice show strong expression of Tfe3 (arrows), which localizes in venous (open arrowheads) and lymphatic endothelial (closed arrowheads) nuclei. **l** Western blot analysis of CD31+ cells isolated from the mesentery at P8. **m** Quantitative PCR analysis of *Tfe3* expression in Pdpn−LYVE1−CD31+ blood endothelial cells (BECs; $n = 4$). **n–u** Images of venous area in tail whole-mounts at P8. Tfe3 proteins are colocalized with RagC (arrows) in control mice, and translocate into nuclei (open arrowheads) in $Flcn^{i\Delta EC}$ mice. **v** Results from two biological HA-TFE3 ChIP-seq in HK-2 cells showing a peak at around 150 bp upstream of the transcription start site of PROX1. **w** Schematic demonstrating location of the PROX1-0.15 kb regulatory element relative to the *PROX1* gene and arrangement of known transcription factor binding sites. **x** Western blotting of HEK293 cells stably transfected with doxycycline-induced TFE3 and quantification of relative intensity of PROX1/HSC70 ($n = 4$). **y** Luciferase reporter assay using HEK293A cells stably expressing dexamethasone-activatable TFE3-glucocorticoid receptor fusion proteins and a 2000 bp fragment of the 5′ region of the human Prox1 gene, with or without a mutation in the E-box sequences. Luciferase activity was quantified using a Dual-Luciferase Reporter Assay System ($n = 3$). Scale bars: 50 μm (**a–d**); 20 μm (**e–j, n, r**); 5 μm (**o–q, s–u**). The comparisons between the averages of the two groups were evaluated using the two-sided Student's *t*-test. **$P < 0.01$; *$P < 0.05$; NS not significant. Data are presented as the mean ± SD. Source data are provided as a Source Data file. Unprocessed original scans of blots are shown in Source Data file.

markers, and the other downregulates Prox1 and remains in the vein[33]. In the lymphovenous valve, Prox1+ endothelial cells line on both sides of the valve, but the mature LEC markers were only expressed on the lymphatic side[4], suggesting LEC-baised VECs have an affinity to bona fide LECs.

We suppose this Flcn function is not related to tumor suppressor activity, but is related to the regulatory function of the stem cell pluripotency and quiescence[21,22,34]. Our data indicate Flcn acts as a gatekeeper limiting the plasticity in committed endothelial cells between venous and lymphatic ones. As lymphatic endothelial cells differentiate from venous ones[6,7], our data fit with the concept that Flcn orchestrates the differentiation status of various cells through nuclear translocation of Tfe3.

In the future, it is quite interesting to analyze the relevance of our data in mice to human BHD pathology, in particular the possibility that bialleic *FLCN* inactivation occurs in VECs of BHD lungs. Otherwise, monoallelic *FLCN* inactivation might be

sufficient to render VECs vulnerable to environmental alterations such as bullae formation caused by epithelial damage. Affected VECs might be readily biased to LECs.

Overall, the data presented herein identify the fundamental molecular mechanism that maintains separation of the blood and lymphatic vascular systems. These findings may provide the foundation for the development of novel therapeutic approaches for human lymphatic disorders as well as to prevent cancer metastasis.

## Methods

**Mouse and matings.** Animal experiments were approved by the regional animal study committees (Keio University) and were performed in accordance with the Guidelines of Keio University for Animal and Recombinant DNA experiments. The *Tie2–Cre*[24], *Prox1-Cre*[ERT27], *Prox1-flox*[35], *Pgc1α-flox*[36], *Tfe3*[+/−37], *CAG-LSL-EGFP*[38], *Cdh5-BAC-Cre*[ERT239], and *Flcn−flox*[40] mice have been described previously. *mTor-flox* mice were obtained from the Jackson Laboratory (Stock No: 011009). All mice were crossed with C57BL/6 J mice more than eight times and

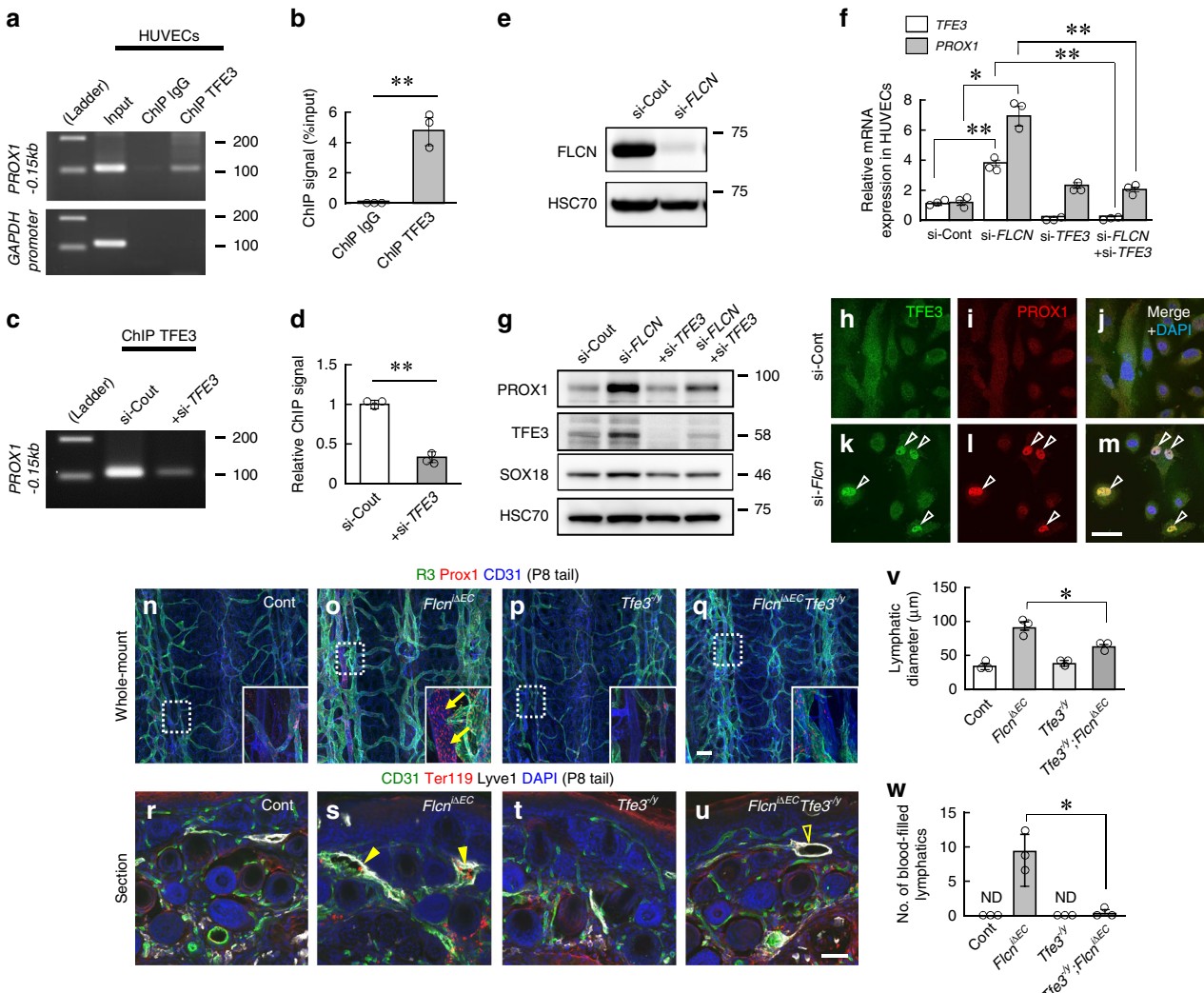

**Fig. 6 Flcn suppresses Prox1 expression through Tfe3. a, b** ChIP analysis of HUVECs to measure/quantify binding of endogenous TFE3 to PROX1-0.15 kb (*n* = 3). **c–d** ChIP analysis of HUVECs treated with or without si-*TFE3* and quantification (*n* = 3). **e** Western blotting of HUVECs treated with or without si-*FLCN*. **f** Quantitative PCR analysis (*n* = 3) of HUVECs treated with si-*FLCN* with or without si-*TFE3*. **g** Western blotting of HUVECs treated with si-*FLCN* with or without si-*TFE3*. **h–m** Immunocytochemistry of HUVECs. PROX1+ cells largely correspond to TFE3+ cells (arrowheads). **n–w** Immunohistochemical analysis of tail whole mounts (**n–q**) or sections (**r–u**) at P8 and quantification (*n* = 3). Venous Prox1 expression (arrows) and blood-filled lymphatics (closed arrowheads) are detected in *Flcn^iΔEC* mice. Hyperplastic LVs in *Flcn^iΔEC Tfe3^-/y* are not blood-filled (open arrowhead). Scale bars: 200 μm (**n–q**); 50 μm (**h–m, r–u**). The comparisons between the averages of the two groups were evaluated using the two-sided Student's *t*-test. **\*\****P* < 0.01; \**P* < 0.05; ND not detected. Data are presented as the mean ± SD. Source data are provided as a Source Data file. Unprocessed original scans of blots are shown in Source Data file.

maintained. For analysis of phenotypes, embryos and neonates of both sexes were used. For neonatal experiments of the tamoxifen-inducible expression of Cre, 4-hydroxytamoxifen (40 μg) was subcutaneously injected at P2, P3.5, P5, and P6.5. To create mice harboring the *Tie2*–Cre-specific deletion, a *Cre^het Flcn^flox/+* male was mated with *Flcn^flox/flox* females to avoid germline expression of *Tie2*–Cre. For inducible double knockout of *Flcn* and *Tfe3* in endothelial cells, a *Cdh5-BAC-Cre^ERT2het Flcn^flox/flox* male was crossed with *Flcn^flox/+ Tfe3^+/-* females. As *Tfe3* is an X-linked gene, male pups with one mutant allele (*Tfe3^-/y*) were used for Tfe3-null mice. Otherwise, both sexes were used for parental Cre/Cre^ERT2.

**Generation of R26-LSL-Prox1 mice.** A mouse Prox1 coding sequence with 5' FLAG-tag was obtained from the pcDNA3-FLAG-Prox1 vector (kindly provided by Dr. Tetsuro Watabe, Tokyo Medical and Dental University) using restriction enzymes EcoRI and XhoI. This FLAG-Prox1 sequence was inserted into the pAi9 vector (Addgene #22799) via the FseI restriction sites to replace the tdTomato sequence. PGK/DTA was removed from this modified pAi9 vector (hereafter *R26-LSL-Prox1*) using KpnI and AvrII. This construct was knocked into the mouse Rosa26 locus using the CRISPR/Cas9 method. Briefly, the beginning and end of the long and short Rosa26 homologous arms were mapped by aligning with the mouse genome database. A CRISPR sgRNA (CGCCCATCTTCTAGAAAGAC) was designed to cut the Rosa26 locus between the long and short homologous arms.

The R26-LSL-Prox1 construct (10 ng/μl) was co-microinjected along with Cas9 mRNA (50 ng/μl) and sgRNA (20 ng/μl) into the pronuclei of fertilized mouse eggs. After culturing the injected embryos overnight in M2 medium at 37 °C/5% $CO_2$, embryos that had reached the 2-cell stage of development were implanted into the oviducts of pseudopregnant foster mothers. Mice born to the foster mothers were genotyped by PCR using Flox primers Prox1–Fw, 5'-TGAGTCCTTAGAC TTGACTCG-3' and Prox1-Rv, 5'-CACGTCCGAGAAGTAGGTC-3'. The PCR comprised 34 cycles as follows: 30 sec at 95 °C, 1 min at 58 °C, 30 sec at 72 °C, followed by 5 min of terminal elongation at 72 °C. Correct homologous recombination of the 5' homology arm of the targeting vector into Rosa26 was confirmed by PCR using primers F1 and R1: F1, 5'-CGCCTAAAGAAGAGGCTGTG-3'; R1, 5'-ATGGGGGAGAGTGAAGCAGAA-3'. This PCR consisted of 34 cycles as follows: 30 sec at 94 °C, 1 min at 62.5 °C, 2 min at 65 °C, followed by 10 min of terminal elongation at 65 °C. Homologous recombination of the 3' vector arm was confirmed by PCR using primers F2 and R2: F2, 5'-AGGGGATCCGCTGTAAGTCT -3'; R2, 5'-CAAGCACTGTCCTGTCCTCA-3'. This PCR consisted of 34 cycles as follows: 30 sec at 94 °C, 45 sec at 62.5 °C, 4.5 min at 65 °C, followed by 10 min of terminal elongation at 65 °C. PCR reactions confirming correct homologous recombination were carried out using a LongAmp Taq PCR Kit (NEB, Cat# E5200S), according to the manufacturer's protocol. The mice can be obtained from Dr. Yoh-suke Mukouyama (mukoyamay@nhlbi.nih.gov).

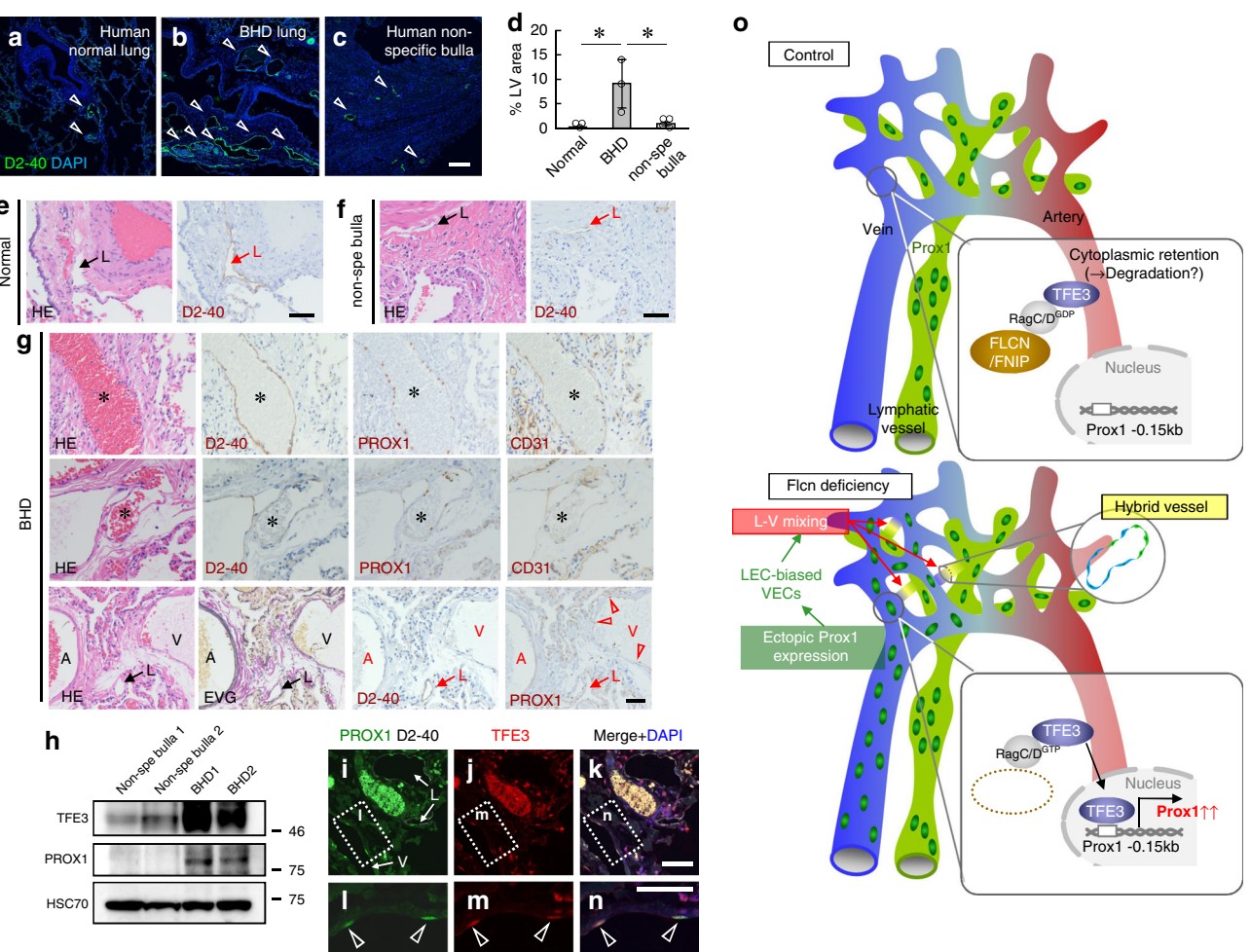

**Fig. 7 FLCN haploinsufficiency causes lymphatic blood filling in humans. a–d** Immunohistochemical analysis and quantification ($n = 3$ in each group). Lungs from BHD patients show increased LVs (arrowheads). **e–g** Immunohistochemical analysis of serial sections. Lungs from BHD patients show blood-filled LVs (asterisks), and Prox1 expression in veins (V; arrowheads) as well as in LVs (L). **h** Western blot analysis of human lung samples. **i–n** Immunofluorescence analysis. Lungs from BHD patients show expression of Prox1 and TFE3 in D2-40⁻ veins (V; arrowheads) as well as in LVs (L). **o** Proposed model for the separation of blood and lymphatic systems. Scale bars: 200 μm (**a–c**); 50 μm (**e–g, i–n**). The comparisons between the averages of the two groups were evaluated using the two-sided Student's t-test. *$P < 0.05$. Data are presented as the mean ± SD. Source data are provided as a Source Data file. Unprocessed original scans of blots are shown in Source Data file.

**Preparation of whole-mount and section tissues**. For preparation of whole-mount tail samples, the distal half of the tail was cutoff and an incision made in the ventral side with a scalpel. The coccyx and muscles were surgically removed and then the samples were flattened for 10 min in 4% paraformaldehyde (PFA) in PBS. To prepare whole-mount skin samples from embryos, cutaneous tissues were removed from underlying muscles and flattened for 10 min in 4% PFA/PBS. For preparation of whole-mount mesenteric samples, intestines were unfolded 'to complete the wheel', according to a previous paper[41]. Isolated tails, skins, and mesenteries were re-fixed overnight before staining with antibodies. For section specimens, tissues were dissected out from mice and fixed overnight in 4% PFA in PBS. All whole-mount or section tissues were stained as described below.

**Immunostaining**. Immunohistochemical staining of whole-mount samples and tissue sections was performed as described previously[42]. The primary monoclonal antibodies used were hamster anti-mouse CD31 (Chemicon, Temecula, CA, USA, MAB1398Z; 1:1,000), rat anti-mouse CD31 (BD Biosciences; Franklin Lakes, NJ, USA, MEC13.3; 1:500), anti-mouse CD45 (BD Biosciences; 550539; 1:500), anti-COUP-TFII (Perseus Proteomics; Tokyo, Japan, H7147; 1:100), anti-Endomucin (Santa Cruz; Santa Cruz, CA, USA; sc-65495; 1:500), anti-Ter119 (R&D Systems, Minneapolis, MN, USA, MAB1125; 1:1,000), and anti-D2-40 (Nichirei, Tokyo, Japan, 413451; 1:1). The primary polyclonal antibodies used were as follows: GFP-Alexa Fluor 488-conjugated (Molecular Probes, Eugene, OR, USA, A21311; 1:500), rabbit LYVE-1 (RELIATech, Braunschweig, Germany; 103-PA50AG; 1:1000), goat LYVE-1 (R&D systems; AF2125; 1:200), Vegfr3 (R&D; AF743; 1:1000), mouse Prox1 (Angiobio; San Diego, CA, USA, 11-002 P; 1:500), human Prox1 (R&D systems; AF2727; 1:500), Podoplanin (MBL; Woburn, MA, USA; D190-3; 1:500),

Tfe3 (Sigma–Aldrich, Saint Louis, MO, USA, HPA023881; 1:500), Cx40 (Alpha Diagnostic; San Antonio, TX, USA; 1:500), Erg (Abcam; Cambridge, UK, ab92513; 1:2,000), Cldn11 (Abcam; ab53041; 1:1,000), vWF (Abcam; ab6994; 1:500), and human CD31 (Agilent Technologies, Santa Clara, CA, USA, 1:50). Secondary antibodies used were Alexa Fluor 488-conjugated IgGs (Molecular Probes, A11034, A11006, A11055; 1:500) or Cy3/Cy5 DyLight549/DyeLight649-conjugated IgGs (Jackson ImmunoResearch, West Grove, PA, USA, 711-165-152, 112-165-167, 127-165-160, 711-605-152, 112-605-167, 127-605-160; 1:500). For nuclear staining, specimens were treated with 4',6-diamidino-2-phenylindole (DAPI; Molecular Probes, D-1306). For double labeling with primary rabbit antibodies, the conjugated Fab fragments method was used (https://www.jacksonimmuno.com/technical/products/protocols/double-labeling-same-species-primary/example-a). Briefly, an excess amount of Alexa Fluor 488-conjugated AffiniPure Fab fragment donkey anti-rabbit IgG (Jackson ImmunoResearch, 711-547-003; 1:100) was applied to achieve effective blocking of the first primary antibody. To analyze cell proliferation in vivo, an EdU incorporation assay using Click-iT EdU Imaging Kits (Invitrogen) was performed according to the manufacturer's instructions. Briefly, 50 μl of EdU dissolved in DMSO/PBS (final concentration, 0.5 mg/ml) was injected intraperitoneally into P8 mice 6 h before sacrifice.

**Angiography and lymphangiography**. For angiography, 50 μl of fluorescent tomato lectin (1 mg/ml; Vector Laboratories; Burlingame, CA, USA) in PBS was injected into the left cardiac ventricle and allowed to circulate for 3 min. For lymphangiography, FITC-dextran (10 mg/ml; MW 2000 kDa; Sigma–Aldrich) in PBS (20 μl) was injected into the subcutaneous tissues of both hindlimbs. TDs were imaged 2 h later.

**Cell culture**. HUVECs were cultured in EGM-2 medium (Cambrex, East Rutherford, NJ). HEK293 cells were cultured in DMEM supplemented with 10% fetal bovine serum (FBS). HK-2, a human kidney proximal tubule cell line, which expresses hemagglutinin (HA)-TFE3 in a doxycycline-dependent manner, was generated by two sequential lentiviral transductions using a Tet-On 3 G System (Clontech) and cultured in Advanced DMEM/F12 medium with 1.5% Tetracycline-Free FBS (Clontech), penicillin−streptomycin (100 U/ml), and selection antibiotics, 2 μg/ml blasticidin S and 0.8 μg/ml puromycin. For RNA interference, HUVECs were washed once with OptiMEM (Life Technologies) and transfected with 40 nM siRNA Duplex (Sigma–Aldrich) using 6 μl/ml Lipofectamine RNAiMAX (Invitrogen) in OptiMEM according to the manufacturer's instructions. After 24 h, the transfection medium was removed, and complete culture medium was added. Cells were cultured for a further 96 h and subsequently a second transfection was performed before each experiment. The FlexiTube GeneSolution was used for *FLCN* (SI03048402, QIAGEN, Hilden, Germany) and ON-TARGETplus siRNA for *TFE3* (L-009363-00-0005, Dharmacon, Cambridge, UK). As a control, a siRNA-negative control duplex oligonucleotide (ThermoFisher Waltham, MA) was used.

**Cell isolation using Dynabeads**. The mesenteric tissues were incubated for 30 min at 37 °C in DMEM containing 1% collagenase D (from *Clostridium histolyticum*; Sigma–Aldrich, Saint Louis, MO), 1 U/ml dispase (ThermoFisher) and 1 U/ml DNase (Invitrogen, Carlsbad, CA) before cells were dissociated by gentle trituration. Cells were isolated using Dynabeads (Veritas; Tokyo, Japan), according to the manufacturer's instructions. To isolate LECs with Lyve1+ macrophages, dissociated cells were incubated simultaneously with an anti-Pdpn antibody (MBL) preconjugated to Dynabeads M-280 anti-Rabbit IgG (DB11203; Veritas) and an anti-LYVE-1 antibody (RELIATech) preconjugated to Dynabeads M-450 anti-Rat IgG (DB11035; Veritas); cells were positively selected (a mix of Pdpn+Lyve1−, Pdpn-Lyve1+ and Pdpn+Lyve1+ cells). To isolate BECs, Pdpn-LYVE1- cells were subsequently incubated with an anti-CD31 antibody (BD Biosciences) preconjugated to Dynabeads M-450 anti-Rat IgG, and positively selected as Pdpn-LYVE1-CD31+ cells. For single-cell RNA sequencing, CD31+ cells were isolated using a biotinylated CD31 antibody (BioLegend, San Diego, CA, USA, BL102504) preconjugated to magnetic beads using a Dynal CELLection Biotin Binder Kit (DB11533; Veritas). The beads were then dissociated from isolated cells using DNase to cut the DNA linker between antibody and beads according to the manufacturer's instructions.

**Single-cell RNA sequencing**. At P8, CD31+ cells were isolated from the mesentery of three mice per genotype using Dynabeads (Veritas). Libraries were constructed using a commercial microdroplet-based platform, Chromium Single Cell 3′ v3 (10×Genomics, Pleasanton, CA) according to the manufacturer's instructions. A HiSeq3000 platform (Illumina, San Diego, CA) was used to generate 28 and 91 base paired-end reads. RNA-seq tags from the Chromium experiments were aligned using Cell Ranger 3.0 software (10×Genomics). Using Cell Ranger, sequences with low quality and PCR duplicates were removed. The scRNA-seq data were processed and analyzed by Seurat package v3.5.2[43]. UMAP was used for the dimensionality reduction that captures both local and global structure in scRNA-seq data. Clusters were visualized in two dimensions as feature or violin plots, and annotated based on the expression of canonical endothelial subtype markers. Clustering itself does not include any supervision such as excluding specific genes or cell populations, but the determination of cell types such as LECs, VECs, AECs was arbitrarily conducted based on the known expression profiles of marker genes. Hierarchical clustering was performed by the Euclidean distance and group average method. Gene expression was averaged in each cluster and the correlation matrix calculated by Pearson correlations. The heatmap was created using the pandas v0.22.0, seaborn v0.8 package. To construct the lists of genes up- or downregulated in LEC-biased VECs, differentially expressed genes were extracted using the Wilcoxon Rank Sum test in the Seurat package. To select genes characterizing the cluster of LEC-biased VECs, genes differentially expressed when compared with all other clusters were extracted using the Seurat package.

**Quantitative real-time RT-PCR analysis**. Total RNA was prepared from cultured cells or mouse blood endothelial cells (Pdpn-LYVE1-CD31+) isolated from the mesentery using Dynabeads as described above. Reverse transcription was performed using Superscript II (Invitrogen). Quantitative PCR assays were performed on an ABI 7500 Fast Real-Time PCR System using TaqMan Fast Universal PCR master mix (Applied Biosystems) and TaqMan Gene Expression Assay Mix with mouse *Tfe3* (Mm00552681_m1), human *PROX1* (Hs00160463_m1), or *TFE3* (Hs00232406_m1). A mouse *β-Actin* (Mm00607939_s1) or a human *β-ACTIN* (Hs00357333_gl) assay mix served as an endogenous control. Data were analyzed using 7500 Fast System SDS Software 1.3.1 (Applied Biosystems). Each experiment was performed with four replicates from each sample and the results were averaged.

**Genomic PCR and RT-PCR analysis**. Detection of *Flcn* mutant alleles by genomic PCR was performed using three primer sets to amplify floxed (292 bp PCR product) and deleted (392 bp PCR product) *Flcn* alleles: P1, 5′-GTTGTCTGGAGTGCTACTTA

GTCAGG-3′; P2, 5′ -CAACACCCCAGCATCCAG-3′; and P3, 5′-CAGCTCCCTC TACCCAGACA-3′. All primer sequences are listed in Supplementary Data 3.

**Confocal microscopy**. Fluorescent images were obtained using a confocal laser scanning microscope (FV1000; Olympus, Tokyo, Japan). Quantification of cells or parameters of interest was conducted in a 1250 × 1250 μm field of view for vessel density and %LV area, and a 200 × 200 μm field of view for counting EdU+ cells using FV10-ASW 3.0 Viewer (Olympus). To quantify the fluorescent intensity of indicated antibodies (Tfe3, CD31, Pdpn, Vegfr3, and Lyve1) using the Image J software (NIH, Bethesda, MD, USA), we measured the "Mean" values in randomly chosen four 50 × 50 μm fields of view in each image and were averaged. Blood-filled lymphatics were counted as the number of cross sections of LV tubes containing one or more erythrocyte in the entire embryo or tail section in scanned images. To get the serial images for z-stack slices, multiple slices horizontally imaged from the same field of view at 0.3-μm intervals were captured.

**Western blotting**. Western blot analysis of cultured cells or human lung samples was performed as described elsewhere[33]. The primary antibodies used were against TFE3 (Sigma–Aldrich, HPA023881; 1:1,000), HA (Sigma–Aldrich, H6908; 1:2,000), PROX1 (Merck, 07-537; 1:1,000), SOX18 (Santa Cruz, sc-166025; 1:500), Histone H3 (Cell Signaling, 9715 S; 1:2,000), FLCN (Cell Signaling, 3697, 1/1000), and HSC70 (Santa Cruz, sc-7298; 1:500).

**ChIP sequencing analysis**. HA-TFE3 inducible HK-2 cell lines were cultured with 200 ng/ml doxycycline for 24 h, and cross-linked with 1% formaldehyde at room temperature for 5 min followed by incubation with 125 mM glycine. Nuclei preparation, chromatin digestion by micrococcal nuclease, chromatin immunoprecipitation with anti-HA antibody (3F10, Roche), and DNA purification were performed with a SimpleChIP Plus Enzymatic Chromatin IP Kit (Cell Signaling) following the manufacturer's protocol. DNA libraries required for high-throughput sequencing were prepared from DNA obtained after ChIP using the NEBNext Ultra DNA Library Prep Kit and NEBNext Multiplex Oligos for Illumina (New England BioLabs). Multiplexed libraries were subjected to cluster generation and sequencing using a NextSeq 500 Kit (75 cycles) with the NextSeq desktop sequencing system (Illumina). Peak detection was performed using the MACS algorithm (PMID:18798982) in Strand NGS software (Strand Life Sciences). HA-TFE3 binding was identified by the significant enrichment of each signal over input-DNA peaks at a $P$ cutoff value of $10^{-5}$. Visualization of ChIP-seq data was performed using Strand NGS software.

**ChIP assay**. Approximately $1 \times 10^7$ HUVECs were fixed in 1% formalin for use in ChIP assays. The fixed cells were sonicated to obtain fragmented chromatin between 200 and 1000 bp in length. An aliquot of sonicated chromatin was put aside to be the input control. ChIP assays were performed using ChIP-IT Express (Active Motif, Tokyo, Japan) with 0.3 mg/ml of anti-TFE3 antibody (1:100, Sigma–Aldrich, HPA023881) or Rabbit IgG as a control. Primers used to amplify a 110 bp product from the human PROX1-0.15 kb region were 5′- TGTGACGTG CAGTCTTCCTG-3′ and 3′-CGGCTGCAATGGTGTATTA-5′. A primer set for detecting the human GAPDH promoter region (5′-CCACATCGCTCAGACAC CAT-3′ and 3′-CCCGCAAGGCTCGTAGAC -5′) was used as a negative control. To quantify the relative abundance of ChIP signals, the intensity of the human PROX1-0.15 kb band was measured using Image J Software (NIH, Bethesda, MD, USA). All primer sequences are listed in Supplementary Data 3.

**Luciferase reporter gene assay**. The 293GPG packaging cells were transfected with pMY-TFE3GR-ires-GFP plasmid vector. HEK293A cells stably expressing TFE3GR fusion proteins were established by infecting the cells with the viral supernatant from 293GPG packaging cells after doxycycline removal. Transduction of >95% GFP-positive cells was confirmed by fluorescent microscopy. A 2000 bp fragment from the 5′ region of the human PROX1 gene was cloned into the pGL4.23 luciferase vector. Two E-box sequences in the PROX1 promoter construct were mutated using a standard mutagenesis protocol and designed PCR primer sets. Reporter plasmids were cotransfected into HEK293A cells along with the phRL-TK vector as an internal control, using PEI-Max (Polysciences, Warrington, PA). The transfection medium was changed to fresh medium either with or without 0.1 μg/ml dexamethasone (DEX) after 24 h, and cultured for 12 h. Luciferase activity was measured and analyzed using the Dual-Luciferase Reporter Assay System, according to the manufacturer's protocols (Promega, Madison, WI).

**Human samples**. Patients who underwent surgical removal of pulmonary cysts were enrolled in this study based on written informed consents for histological and biochemical analyses. All patients received genetic counseling and were diagnosed as BHD. The study was approved by the Institutional Review Board of Yokohama City University (approval number, A110929001) and Keio University School of Medicine Ethics Committee (approval number, 20170369). The human study was conducted in compliance with the Japanese regulation, Ethical Guidelines for Medical and Health Research Involving Human Subjects, in addition to

internationally established principles of the Declaration of Helsinki. Resected lung tissues ($n = 3$) were fixed with 10% buffered formalin and embedded in paraffin. Cystic lesions and surrounding noncystic pulmonary tissues from patients with BHD ($n = 3$) and spontaneous bullae ($n = 3$) were snap-frozen and stored in liquid nitrogen prior to western blotting. Four-micrometer thick paraffin sections were subjected to immunohistochemistry. After deparaffinization and rehydration, sections were autoclaved at 121 °C for 15 min.

**Statistics and reproducibility**. Results are expressed as the mean ± standard deviation (SD). The comparisons between the averages of the two groups were evaluated using the two-tailed Student's $t$-test. $P$ values of <0.05 were considered statistically significant. For histological analyses, at least three but typically more independent samples were quantified or qualitatively analyzed with each experimental repeat yielding highly similar results.

**Reporting summary**. Further information on research design is available in the Nature Research Reporting Summary linked to this article.

## Data availability

For scRNA-seq data, raw data will be available in GEO (GSE133512). For ChIP-seq (Fig. 4p), raw data are available in GEO (GSE135490). There is no restriction on data availability. Source data are provided with this paper.

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

## Acknowledgements

We thank Guillermo Oliver of Northwestern University for kindly providing *Prox1-CreERT3* mice. This work was supported by Grants-in-Aid for Specially Promoted Research from the Ministry of Education, Culture, Sports, Science, and Technology of Japan (22122002, 25713059, 15K15089, 18H05042, 18K19553, 17K15625, 18K16997, 18H02938, 18K19619, and 19K07389), by AMED-PRIME (JP19gm6210017h0001, JP20gm6210017h0002), and by research grants from the following: Inamori Foundation, The Kao Foundation for Arts and Culture, Takeda Science Foundation, Mochida Memorial Foundation, The Mitsubishi Foundation, The Cell Science Research

Foundation, SENSHIN Medical Research Foundation, The Sumitomo Foundation, Daiichi Sankyo Foundation of Life Science, The Naito Foundation, The Uehara Memorial Foundation, The Joint Usage/Research Center Program of the Advanced Medical Research Center, Yokohama City University, and Toray Science Foundation. This research was supported in part by the Intramural Research Program of the NIH, National Cancer Institute, Center for Cancer Research.

## Author contributions

Designed experiments: Y.H., M.B., and Y.K. Performed experiments: I.T-N., Y.H., H.H., K.O., T.A., C.L., X.L., S.F., Y.Suzuki and Y.Satou, and Y.K. Analyzed the data: I.T-N., D.K., M.F., and Y.K. Provided experimental materials: F.I., F.M., H.S., W.L.,Y-S.M., W.M.L., M.H., and M.B. Edited the manuscript: M.B., W.M.L., and M.F. Wrote the manuscript: Y.K. I.T-N. and Y.H. contributed equally to this work.

## Competing interests

The authors declare no competing interests.
