## [Peer Review File · Nature Communications]

REVIEWER COMMENTS

Reviewer #1 (Remarks to the Author):

Tai-Nagara and colleagues showed that inactivation of FLCN in the VEC results in the appearance of lymphatic-biased venous endothelial cells caused by TFE3-mediated ectopic activation of the master lymphatic specification transcription factor Prox1.

Their work consists of a series of in vivo inducible and targeted Flcn deletion experiments in mice, complemented with biochemical proof-of-concept experiments that show convincingly that in the absence of Flcn in venous endothelial cells TFE3 binds to the promoter of Prox1 and in a Prox1-dependent way creates LV with transition phenotype between LV and BV. This results in functional communication between blood and lymphatic system (lectin expt and presence of blood). They also showed that the lymphatic vessels in the lungs of BHD patients exhibit this transitional phenotype in proximity to the formed bullae.

This is an elegant piece of work that ties with the known function of FLCN regulating TFE3. The data are convincing regarding the causative relation of lack of FLCN and vein-lymphatic communication in an experimental animal system.

In my opinion the observations beg the response to a major unanswered question; to what extent this phenomenon relates to the tumor suppressor activity of FLCN or, if this is not the case, which is the other function of FLCN in humans that it highlights. Since this has been engineered in animals what is the relevance to human biology? There is a hint that this phenomenon may relate to bullae formation in the human lungs of the BHD patients. Such a hypothesis would need though more experimental testing either in a transgenic animal or by tissue interrogation.

Specifically, do the ECs of the human BHD lung have bi-allelic inactivation of FLCN? Is this inactivation specific to BHD bullae versus non-specific cyst formation? If this is the case is the loss of wild type allele precede cyst formation (detectable in non cystic parenchyma)? Do lymphatic vessels in fibrofolliculomas or human RCC display the same phenotype?

Minor points

Figure 6C. Needs IgG control in the siFTE3 Chip

Figure 6F indicates that FLCN is not only responsible for FTE3 localization but also controls FTE3 expression

Figure 7. IHC and IF in NORMAL and non-specific bullae lung are required here for specificity.

Reviewer #2 (Remarks to the Author):

Using a tamoxifen-inducible, endothelial specific FLCN knockout mouse model, the authors describe an unanticipated function of the tumor suppressor FLCN in lymphatic – venous differentiation. Endothelial FLCN deletion after birth induces Prox1 expression in venous endothelial cells, lymphatic-venous shunts, and early death in mice. The phenotype is presented in a detailed and convincing way, with largely excellent microscopy. In comparison, the investigation of the underlying mechanism is somewhat falling behind.

Major comments:

1. Endothelial FLCN deletion results in highly varying effects on the circulatory system, from no phenotype in arterial ECs to Prox1 induction in venous ECs to hyperproliferation in lymphatic ECs. What is the molecular basis for these divergent roles of FLCN? How is FLCN and its effector Tfe3 expressed in these different vascular beds?

2. Do the FLCN Δ EC recapitulate the phenotype of the Tie2Cre-driven FLCN knockout mouse if tamoxifen is given during embryonic development? Is venous Prox1-induction and venous-lymphatic shunt formation also occurring when tamoxifen is applied to adult animals (as in Suppl. Figure 1)?
3. How do the lympho-venous shunts form after postnatal tamoxifen application in FLCN Δ EC mice? Is there sprouting of initially separated vessels towards each other, or do these vessels form wrongly from the beginning? A more detailed time series of this phenotype from the last tamoxifen injection until P8 would be helpful.
4. Figure 4 f-j shows a normalization of the phenotypes of Flcni Δ EC mice after ectopic Prox1 deletion. What was the effect of Prox1 deletion on the function and/or differentiation of the lymphatic vascular system?

Minor:

1. Please carefully check all figure legends. Some panels lack a legend, for others the labeling in the figure and the legend do not match.
2. Supplementary Figure 1: The tumor study appears circumstantial (since it only shows that there are less (no) peri-tumoral LVs in control mice compared to FLCN Δ EC mice) and is not well presented. Growth curves should be shown instead of the endpoint diameter. Why was the histological analysis done at day 12 and not at day 22? Which vessels (BV or LV) were assessed in S1d? Y-axis label in S1D seems wrong ("% would refer to an area fraction, but "vessel density" to the number of vessels / area).
3. Line 161: These data (Cldn11 upregulation and vWF downregulation) should be shown.
4. Line 172: Fig reference should be Fig 4c-e", instead of 3c-e".
5. Line 189: Please show these data.
6. Figure 5: Cytoplasmic vs. nuclear localization cannot be the only reason for the dramatically increased nuclear Tfe3 staining in Figs. 5a-d. If the transcription of Tfe3 is not changed upon FLCN deletion, there might be a post-transcriptional mechanism inducing Tfe3 protein levels. This should be discussed better. Furthermore, the images in 5n-u are underwhelming. Clearer microscopy images should be provided.
7. Figure 5x: the slow kinetic of Prox1 induction compared to Tfe3 induction suggests rather an indirect regulation of Prox1 expression by Tfe3. This should be discussed.
8. Fig. 6e is trivial and could be removed.
9. The discussion is too short and fails to accurately relate the findings of this study with the current literature context. For example, a deeper discussion of prox1-expressing venous ECs (which have been described before and have been shown not to differentiate into lymphatic ECs) and the molecular regulation of Prox1 expression would be helpful.

Reviewer #3 (Remarks to the Author):

In this study, Tai-Nagara et al. investigated the mechanism allowing the segregation of lymphatic

and blood vessel systems. They discovered that the gene Flcn had an important function in that process. Using various mouse models, they showed that loss of this gene specifically in the vascular system induced the ectopic expression of lymphatic vessel genes in veins. This change in vein properties disrupted the segregation between the lymphatic and blood circulation systems. The authors revealed that loss of Flcn was allowing the transcription factor TFE3 to activate its target gene Prox1, a master regulator of lymphatic endothelial cell fate. Loss of TFE3 in the context of a conditional Flcn knockout in the vascular system rescued the negative consequences of Flcn loss of function. Finally, they went on to show that human patients with Flcn haploinsufficiency had abnormalities in their lymphatic system suggesting an evolutionary conserved Flcn role in blood vessels.

This work addressed a very important question in the field. Besides, this study is very thorough making use of an impressive number of mouse models. Moreover, they also performed single-cell RNA sequencing and Chip sequencing experiments to have a molecular understanding of Flcn function in blood vessels. Overall, this study is very convincing. I have nonetheless some comments that would need to be addressed before considering publication.

1) The authors started their manuscript stating the phenotype of the Flcn knockout mice which has nothing to do with the vascular system. It is unclear why they decided to study Flcn. What was their motivation? Did they find evidence of its expression in the blood or lymphatic vessels?

2) One of the consequences of Flcn loss of activity is the widening of lymphatic vessels (lymphatic hyperplasia). However, the authors did not explain this phenotype. Why are the lymphatic vessels becoming bigger upon Flcn loss of function?

3) In lines 117-120, they mention some work with a mouse cancer model. It is completely out of sync with the rest of the study. Is it necessary to keep it?

4) They chose to use the mesentery for their sc-RNA-seq study. Why this particular tissue? The sc-RNA-seq analysis was well done and informative. The identification and characterisation of the LEC biased VEC population was very convincing. These cells were found exclusively in the Flcni Δ EC mice. However, the entire VEC cluster was composed of cells coming from wild type and Flcn mutant cells. Did the authors find any difference between them? If not, why? Could it be that the Flcn gene was not lost in the cells coming from Flcni Δ EC mesentery? Another question is about the LEC1 cluster. Why is it almost entirely composed of Flcni Δ EC cells?

In figure 3k, the colour scheme used for the heatmap is counterintuitive. The darker colour means here low expression while the light one means high expression. This is confusing. I recommend to change the colour scale.

5) In lines 187-189, the authors mention that overexpression of Prox1 in embryonic endothelial cells resulted in blood-filled lymphatics. It is an interesting result and it will be informative for the reader to see it.

6) In lines 195-196, the authors introduced Tfe3 by writing "well known as a downstream effector of Flcn signaling". This statement is misleading. It gives the impression that when Flcn is present, Tfe3 is active whereas Flcn physical interaction with Tfe3 actually prevents the latter to drive transcription.

7) In lines 199-200, they wrote "The specificity of this staining was confirmed using Tfe3 deficient mice (data not shown)." Considering how important Tfe3 is for the rest of the study, I recommend the authors to include this staining control in supplementary information.

8) In line 206, they introduced their Chip-Seq experiment. It is unclear why they chose to do this experiment in the HK-2 cell line which is not endothelial. Why not use the HUVEC? Or another endothelial cell line?

9) In figure 6, they performed a number of experiments with siRNAs. However, they did not show anywhere that the Flcn siRNA was effectively reducing the expression of the corresponding protein.

10) The discussion is a bit short. Although this study is focused on the vascular system, it would be interesting to compare the phenotype they observed in endothelial cells with the role of Flcn in other systems such as ESC (Betschinger et al. 2013, PMID: 23582324) or hematopoietic stem cells (Baba et al. 2016, PMID: 27095138). That would help the reader to have a broader view on the function of Flcn in different cellular contexts.

Reviewer #4 (Remarks to the Author):

The authors studied a molecular mechanism that appears to maintain a separation of blood and lymphatic vessels. They studied in a mouse model the genetic deficiency of folliculin, a tumor suppressor, leading to misconnection of blood and lymphatic vessels in mice and humans. Absence of folliculin resulted in the appearance of lymphatic-biased venous endothelial cells caused by ectopic expression of Prox 1.

Interestingly, nuclear translocation of the basic helix-loop-helix (bHLH) transcription factor Transcription Factor E3 (TFE3), binding to an unprecedented regulatory element of Prox1, enhanced its venous expression. The authors conclude that folliculin acts as a gatekeeper maintaining the separation of blood and lymphatic vessels by limiting the plasticity of committed endothelial cells.

Clearly, the presented data are novel and convincing and will be of great interest to others in the community and the wider field. In addition, the statistical analysis is appropriate and the level of details of the methods provided is sufficient to allow a researcher to reproduce the work. All in all, I suggest to accept the manuscript in its present form.

Comments of Referee #1 (blue) and our responses (black):

Tai-Nagara and colleagues showed that inactivation of FLCN in the VEC results in the appearance of lymphatic-biased venous endothelial cells caused by TFE3-mediated ectopic activation of the master lymphatic specification transcription factor Prox1.

Their work consists of a series of in vivo inducible and targeted Flcn deletion experiments in mice, complemented with biochemical proof-of-concept experiments that show convincingly that in the absence of Flcn in venous endothelial cells TFE3 binds to the promoter of Prox1 and in a Prox1-dependent way creates LV with transition phenotype between LV and BV. This results in functional communication between blood and lymphatic system (lectin expt and presence of blood). They also showed that the lymphatic vessels in the lungs of BHD patients exhibit this transitional phenotype in proximity to the formed bullae. This is an elegant piece of work that ties with the known function of FLCN regulating TFE3. The data are convincing regarding the causative relation of lack of FLCN and vein-lymphatic communication in an experimental animal system.

Thank you very much for evaluating the importance of our paper and overviewing each data from the viewpoint of the overall significance. Also, we sincerely appreciate your expert comments by which we could greatly improve the paper. Our point-by-point responses are listed below:

In my opinion the observations beg the response to a major unanswered question; to what extent this phenomenon relates to the tumor suppressor activity of FLCN or, if this is not the case, which is the other function of FLCN in humans that it highlights.

Thank you for this insightful suggestion. We suppose this FLCN function is not related to tumor suppressor activity, but is somewhat related to the regulatory function of the stem cell pluripotency (Tsun et al., *Cell* 2013; Villegas et al., *Cell Stem Cell* 2018). Our data indicate FLCN acts as a gatekeeper limiting the plasticity in committed endothelial cells between venous and lymphatic ones. As these two endothelial cells have the same origin, our data fit with the concept that FLCN orchestrates the differentiation status of various cells through nuclear translocation of TFE3. This point is discussed in **lines 282-287**.

Since this has been engineered in animals what is the relevance to human biology?

There is a hint that this phenomenon may relate to bullae formation in the human lungs of the BHD patients. Such a hypothesis would need though more experimental testing either in a transgenic animal or by tissue interrogation. Specifically, do the ECs of the human BHD lung have bi-allelic inactivation of *FLCN*? Is this inactivation specific to BHD bullae versus non-specific cyst formation? If this is the case is the loss of wild type allele precede cyst formation (detectable in non cystic parenchyma)?

It is quite interesting to analyze the relevance of our data in mice to human BHD pathology, in particular the possibility that that bi-allelic *FLCN* inactivation occurs in VECs of BHD lungs. Otherwise, mono-allelic *FLCN* inactivation might be sufficient to render VECs vulnerable to environmental alterations such as bullae formation caused by epithelial damage. Affected VECs might be readily biased to LECs as shown in **Fig. 7**. To test these hypotheses, we have to obtain live lung samples from BHD patients, and conduct FACS sorting to isolate endothelial cells. In our institutes, genomic analysis of BHD is done after surgical removal of bullae and pathological examination. Therefore, we have to perform *de novo* lung biopsy of BHD patients just for our research. Ethically, we assume that this experiment is extremely difficult. Laser capture microdissection on fixed tissues might be helpful to circumvent this ethical issue. However, we suppose it is nearly impossible to isolate venous endothelial cells, monolayer squamous epithelium lining the venous lumen. Even if this is doable, it will take more than one year to get institutional approval and informed consent from BHD patients. Taking these situations we discuss this point carefully (**lines 288-293**), and would like to test these interesting hypotheses in our future study.

Minor points

Figure 6C. Needs IgG control in the siTFE3 Chip

Thank you for this suggestion. This request might be related to the specificity of TFE3 antibodies we used. If so, the relevant data is shown in **Fig. 6a** (Please see the “ChIP IgG” lane). Our aim of **Fig. 6c** was to support the result of **Fig. 6a** by excluding the possibility that co-precipitation of Prox1 promoter is mediated by non-specific binding of anti-TFE3 antibody to proteins other than TFE3. Otherwise, if you are questioning about the cause of the weak band in the “si-TFE3” lane of **Fig. 6c**, it represents the leftover TFE3 proteins after knockdown, but not non-specific binding of TFE3 antibodies. Indeed, we could detect a small amount of TFE3 mRNA remaining after si-*TFE3* treatment (**Response Figure 1 below: extracted from the data shown in Fig.**

6f).

Response Figure 1

Figure 6F indicates that FLCN is not only responsible for FTE3 localization but also controls FTE3 expression

Thank you very much for pointing out this important point. In terms of the effect of *FLCN* deletion on *TFE3* expression, we noticed a discrepancy between cultured cells and mice; HUVECs treated with si-*FLCN* show significantly increased *TFE3* transcription (**Fig. 6f**), while endothelial cells in *Flcn*^{i EC} show unchanged *Tfe3* expression (**Fig. 5m**). This may be explained at least in part by the feedback loop between FLCN and TFE3 (Endoh et al., *Cell Rep* 2020), which is disrupted by FLCN loss. We assume this feedback loop might be masked by paracrine factors from surrounding cells or some molecular mechanisms like other bHLH transcription factors *in vivo*. Supporting this assumption, we show the violin plot for *Tfe3* in scRNA-seq. The data show *Tfe3* is indeed expressed in LEC-biased VECs but the level is not dramatically increased compared to VECs (**Response Figure 2 below**). We suppose the increased level of Tfe3 proteins in the absence of *Flcn* is largely dependent on non-transcriptional regulation like nuclear translocation and lysosomal degradation related to RagGTPases as shown in ES cells (Betschinger et al *Cell* 2013; Villegas et al *Cell Stem Cell* 2018; Mathieu et al *Nat Commun* 2019).

Response Figure 2

Figure 7. IHC and IF in NORMAL and non-specific bullae lung are required here for specificity.

We now conducted the same staining in tissue samples from normal lungs and non-specific bullae (**Fig. 7e-h**). We did not observe blood filling in lymphatic vessels in those samples suggesting this phenomenon, blood filling in lymphatic vessels, is specific to BHD lungs.

Comments of Referee #2 (blue) and our responses (black):

Using a tamoxifen-inducible, endothelial specific FLCN knockout mouse model, the authors describe an unanticipated function of the tumor suppressor FLCN in lymphatic – venous differentiation. Endothelial FLCN deletion after birth induces Prox1 expression in venous endothelial cells, lymphatic-venous shunts, and early death in mice. The phenotype is presented in a detailed and convincing way, with largely excellent microscopy. In comparison, the investigation of the underlying mechanism is somewhat falling behind.

Thank you very much for evaluating our results from the viewpoint of your expertise in lymphangiogenesis. Although we have shown some mechanistic investigation centered on Tfe3 using *in vitro* experiments and concomitant deletion in mice (**Fig. 4-6**), we expanded the time window of our observation and analyzed dynamic changes of phenotypes according to your suggestion. Our point-by-point responses are listed below:

Major comments:

1. Endothelial FLCN deletion results in highly varying effects on the circulatory system, from no phenotype in arterial ECs to Prox1 induction in venous ECs to hyperproliferation in lymphatic ECs. What is the molecular basis for these divergent roles of FLCN? How is FLCN and its effector Tfe3 expressed in these different vascular beds?

Thank you for this insightful comment. We reanalyzed the scRNAseq data, and found the transcription of *Flcn*, *Tfe3*, *Fnip1*, and *RagC*, which are all related to the current molecular cascade, is significantly higher in venous endothelial cells than arterial endothelial cells (**Response Figure 3 below**). This expression tendency may be, at least in part, the determinants for the VEC-inclined phenotype in *Flcn^{i EC}* mice. However we expect epigenetic regulation like chromatin structure and methylation status (reviewed in Swift & Weinstein *Circulation* 2009) as well as lysosomal degradation which highly affects the amount and behavior of Flcn and TFE3 (Villegas et al *Cell Stem Cell* 2018; Betschinger et al *Cell* 2013; Starling et al *EMBO Rep* 2016) might be the most important factor. This point is very interesting, but detailed experiments determining what factor is crucial would be far beyond our study. So we would like to analyze thoroughly in the future study.

Response Figure 3

2. Do the $Flcn^{i\ EC}$ recapitulate the phenotype of the Tie2Cre-driven FLCN knockout mouse if tamoxifen is given during embryonic development? Is venous Prox1-induction and venous-lymphatic shunt formation also occurring when tamoxifen is applied to adult animals (as in Suppl. Figure 1)?

According to your comment, we analyzed $Flcn^{i\ EC}$ embryos at E15.5 three days after tamoxifen administration (**new Supple Fig. 1a-e**), and found they recapitulated the defects of $Tie2-Cre^+ Flcn^{fl/fl}$ embryos. We also analyzed adult $Flcn^{i\ EC}$ mice treated with tamoxifen after weaning (**Supple Fig. 1f-n**). As whole-mount staining was not applicable to adult tails, we examined section specimens. We observed ectopic expression of Prox1 in VECs but did not see lymphatic hyperplasia and blood filling in $Flcn^{i\ EC}$ mice, suggesting some compensation mechanism occurs after weaning and masks the phenotypes seen in their neonatal period. Of note, as shown in **former Supple Fig. 1**, which has been omitted according to another reviewer's recommendation, blood-filled lymphatics appears in pathological settings like cancer even in adult.

3. How do the lympho-venous shunts form after postnatal tamoxifen application in $Flcn^{i\ EC}$ mice? Is there sprouting of initially separated vessels towards each other, or do these vessels form wrongly from the beginning? A more detailed time series of this phenotype from the last tamoxifen injection until P8 would be helpful.

We appreciate this insightful comment. We closely looked at veins and lymphatic vessels in $Flcn^{i\ EC}$ mice. We observed abnormal sprouting from LVs toward veins as

well as from veins toward LVs in *Flcn*^{iAEC} mice (**new Supplementary Fig. 2r–w**). We also detected occasional bridging of endothelial filopodia between LVs and veins in *Flcn*^{iAEC} mice (**new Supplementary Fig. 2w**). Please note that *Vegfr3* expression is upregulated in veins of *Flcn*^{iAEC} mice compared to those in control mice. These abnormal sprouting was never observed in control mice, suggesting that initially separated LVs and lymphatic-biased veins somehow attract each other, eventually forming LV-vein shunts. We recognize these images are suggestive but not conclusive to tell the dynamics associated with shunt formation. In the present paper, we discuss this point as a speculation, and expect a future advance of *in vivo* imaging technology might clarify conclusively how these shunts are formed.

4. Figure 4 f-j shows a normalization of the phenotypes of *Flcn* Δ EC mice after ectopic *Prox1* deletion. What was the effect of *Prox1* deletion on the function and/or differentiation of the lymphatic vascular system?

Thank you very much for pointing out this important point. We carefully checked the phenotype of single knockout of *Prox1* in postnatal endothelial cells (*Cdh5-Cre*^{ERT2+}*Prox1*^{fl/fl}). Although they did not have apparent impairment in overall structures of lymphatic vascular networks, they showed absence of lymphatic valves (**Response Figure 4 below**), in accordance with *Prox1* expression almost exclusive in lymphatic valves at this stage (arrowheads). Absence of lymphatic valves are also seen in *Flcn*;*Prox1*^{iAEC} mice (**Fig. 4s**), but we dared not describe this abnormality considering the main focus of our study.

Response Figure 4

Minor:

1. Please carefully check all figure legends. Some panels lack a legend, for others the labeling in the figure and the legend do not match.

Thank you for carefully checking our draft. We identified a number of errors in legends and corrected all of them.

2. Supplementary Figure 1: The tumor study appears circumstantial (since it only shows that there are less (no) peri-tumoral LVs in control mice compared to FLCNⁱEC mice) and is not well presented. Growth curves should be shown instead of the endpoint diameter. Why was the histological analysis done at day 12 and not at day 22? Which vessels (BV or LV) were assessed in S1d? Y-axis label in S1D seems wrong (“%” would refer to an area fraction, but “vessel density” to the number of vessels / area).

Thank you for this comment. According to another reviewer’s recommendation, we omitted the tumor data, as it is out of sync with the rest of the study.

3. Line 161: These data (Cldn11 upregulation and vWF downregulation) should be shown.

We have now presented immunostaining with Cldn11 and vWF on whole-mount mesenteric samples. In accordance with the scRNA-seq data, the intensity of Cldn11 expression in veins was significantly higher in *Flcn*^{i EC} than in control mice, whereas that of vWF was significantly lower in *Flcn*^{i EC} mice (**new Supple Fig. 4b-d**). These data strengthen the validity of the scRNAseq data.

4. Line 172: Fig reference should be Fig 4c-e”, instead of 3c-e”.

We corrected this type. Thanks!

5. Line 189: Please show these data.

We now show endothelial Prox1-overexpressing (*Prox1*^{iEC-OE}; *Cdh5-Cre*^{ERT2+}*R26-LSL-Prox1*⁺) mice, which upon tamoxifen injection at E13.5, demonstrate lymphatic enlargement and blood-filled lymphatics at E15.5 (**new Fig. 4f-l**; **new Supple Fig. 5**), similar to *Flcn*^{EHC} embryos (**Fig. 1a-l**).

6. Figure 5: Cytoplasmic vs. nuclear localization cannot be the only reason for the dramatically increased nuclear Tfe3 staining in Figs. 5a-d. If the transcription of Tfe3 is not changed upon FLCN deletion, there might be a post-transcriptional mechanism inducing Tfe3 protein levels. This should be discussed better. Furthermore, the images in 5n-u are underwhelming. Clearer microscopy images should be provided.

Thank you for this comment. We suppose the increased level of Tfe3 proteins in the absence of *Flcn* is largely dependent on non-transcriptional regulation like nuclear translocation and lysosomal degradation related to RagGTPases as shown in ES cells (Betschinger et al *Cell* 2013; Villegas et al *Cell Stem Cell* 2018; Mathieu et al *Nat Commun* 2019). We mentioned this point in **lines 210-211**. In addition to **Fig. 5n-u**, we now show serial images for z-stack slices at 0.3 μm intervals (**new Supple Fig. 7m, n**) above and below the single slices shown in **new Supple Fig. 7e-l**. These Z-stack slices show the overlapped substances of RagC/Tfe3 in control mice and nuclear Tfe3 in *Flcn*^{i Δ EC} mice are indeed located in venous endothelial but not perivascular (non-endothelial) cells.

7. Figure 5x: the slow kinetic of Prox1 induction compared to Tfe3 induction suggests rather an indirect regulation of Prox1 expression by Tfe3. This should be discussed.

We did show TFE3 directly binds the enhancer Prox1-0.15kb (**Fig. 5v, w**), and deletion of E-box sequences in PROX1-0.15kb remarkably reduces the PROX1 induction by TFE3 (**Fig. 5y**). These data clearly support the direct regulation of PROX1 expression by TFE3. However, as you noted, we do not exclude definitively the contribution of some indirect effect, intermittent molecules between TFE3 and PROX1. The late induction of PROX1 caused by TFE3 may suggest this possibility. We mentioned this point in **lines 222-223**.

8. Fig. 6e is trivial and could be removed.

According to your recommendation, we removed the **former Fig. 6e**.

9. The discussion is too short and fails to accurately relate the findings of this study with the current literature context. For example, a deeper discussion of prox1-expressing venous ECs (which have been described before and have been shown not to

differentiate into lymphatic ECs) and the molecular regulation of Prox1 expression would be helpful.

Thank you for this recommendation. As the known regulators of Prox1 have been discussed in **Lines 264-271**, we discussed more about the meaning of Prox1 expressing (LEC-biased) VECs in the formation of LV-BV misconnection as (**Lines 272-281**).

Comments of Referee #3 (blue) and our responses (black):

In this study, Tai-Nagara et al. investigated the mechanism allowing the segregation of lymphatic and blood vessel systems. They discovered that the gene *Flcn* had an important function in that process. Using various mouse models, they showed that loss of this gene specifically in the vascular system induced the ectopic expression of lymphatic vessel genes in veins. This change in vein properties disrupted the segregation between the lymphatic and blood circulation systems. The authors revealed that loss of *Flcn* was allowing the transcription factor TFE3 to activate its target gene *Prox1*, a master regulator of lymphatic endothelial cell fate. Loss of TFE3 in the context of a conditional *Flcn* knockout in the vascular system rescued the negative consequences of *Flcn* loss of function. Finally, they went on to show that human patients with *Flcn* haploinsufficiency had abnormalities in their lymphatic system suggesting an evolutionary conserved *Flcn* role in blood vessels.

This work addressed a very important question in the field. Besides, this study is very thorough making use of an impressive number of mouse models. Moreover, they also performed single-cell RNA sequencing and Chip sequencing experiments to have a molecular understanding of *Flcn* function in blood vessels. Overall, this study is very convincing. I have nonetheless some comments that would need to be addressed before considering publication.

Thank you very much for clearly summarizing our results and evaluating the novelty and importance of this paper in the cardiovascular field. We appreciate your professional and constructive advices, because we realize they greatly improved our paper. Our responses to your comments are listed below:

1) The authors started their manuscript stating the phenotype of the *Flcn* knockout mice which has nothing to do with the vascular system. It is unclear why they decided to study *Flcn*. What was their motivation? Did they find evidence of its expression in the blood or lymphatic vessels?

Thank you for this insightful comment. Honestly, the start point of our study is that we just happened to find embryonic lethality of *Tie2-Cre⁺Flcn^{fl/fl}* mice. As one of our strategies to identify important genes, which functions have not been well characterized in the cardiovascular system, we broadly and blindly cross multiple floxed lines with *Tie2-Cre* and first evaluate the lethality. We assume this direct but convenient strategy

efficiently utilizes the nature of cardiovascular system; if something wrong happens, the mice do not survive until birth. This is not the case with other systems like neuronal and hematopoietic systems, but just fit the cardiovascular system. As this kind of statement is not appropriate for being written in scientific papers, we dared not describe in the manuscript.

2) One of the consequences of Flcn loss of activity is the widening of lymphatic vessels (lymphatic hyperplasia). However, the authors did not explain this phenotype. Why are the lymphatic vessels becoming bigger upon Flcn loss of function?

Thank you for pointing out this important issue. As we have shown, the ratio of EdU⁺ in Prox1⁺Pdpr⁺ cells was remarkably higher in *Flcn*^{iAEC} mice (**Fig. 3m, n, q**), suggesting the proliferation of LECs is indeed increased by *Flcn* loss of function. This hyper proliferation of LECs is likely responsible for lymphatic enlargement, and is caused by increased VEGFR3 expression (**Supple Fig. 2m–q**) enhancing ligand responsiveness. This point is discussed this point in the text (**Lines 145-147**). However, considering the length of EdU incorporation, 6 hours, hyperproliferation may not be the only cause of massive lymphatic enlargement; altered fluid pressure may contribute to it, nonetheless we dared not mention this point as is too speculative.

3) In lines 117-120, they mention some work with a mouse cancer model. It is completely out of sync with the rest of the study. Is it necessary to keep it?

Thank you for advice. We omitted the tumor data, as it is relatively out of focus in this study.

4) They chose to use the mesentery for their sc-RNA-seq study. Why this particular tissue? The sc-RNA-seq analysis was well done and informative. The identification and characterisation of the LEC biased VEC population was very convincing. These cells were found exclusively in the *Flcni*ΔEC mice. However, the entire VEC cluster was composed of cells coming from wild type and *Flcn* mutant cells. Did the authors find any difference between them? If not, why? Could it be that the *Flcn* gene was not lost in the cells coming from *Flcni*ΔEC mesentery? Another question is about the LEC1 cluster. Why is it almost entirely composed of *Flcni*ΔEC cells?

In figure 3k, the colour scheme used for the heatmap is counterintuitive. The darker colour means here low expression while the light one means high expression. This is

confusing. I recommend to change the colour scale.

We initially tested the tail skin, lung and mesentery for scRNA-seq study. In our pilot study for the construction of libraries using microdroplet-based platforms, we found the mesentery is the best in terms of the purity, amount, and quality of cDNA. Regarding the difference of the VEC cluster between control and *Flcn^{i EC}* mice, global transcriptional profile indicated those clusters are closely similar (Pearson correlation coefficient: 0.9909) (**Response Fig. 5a below**). Also the expression of specific markers for LEC, AEC, and VEC were comparable (**Response Fig. 5b below**). There might be a threshold of TFE3 activity for converting VECs into LEC-biased VECs.

Response Figure 5

As for your question about the LEC1 cluster please check **Fig. 3e-g**. In *Flcn^{i EC}* mice, the number of LECs are entirely increased, but the ratio of LEC1, LEC2, and LEC3 per all LECs were not changed. We summarize this quantification in **Response Fig. 6 below**.

Response Figure 6

For **Fig. 3k**, we inverted the color scale. Now it seems more understandable. Thanks.

5) In lines 187-189, the authors mention that overexpression of Prox1 in embryonic endothelial cells resulted in blood-filled lymphatics. It is an interesting result and it will be informative for the reader to see it.

We now show endothelial Prox1-overexpressing (*Prox1*^{IEC-OE}; *Cdh5-Cre*^{ERT2+}*R26-LSL-Prox1*⁺) mice, which upon tamoxifen injection at E13.5, demonstrate lymphatic enlargement and blood-filled lymphatics at E15.5 (**new Fig. 4f-l**; **new Supple Fig. 5**), similar to *Flcn*^{EHC} embryos (**Fig. 1a-l**).

6) In lines 195-196, the authors introduced Tfe3 by writing “well known as a downstream effector of Flcn signaling”. This statement is misleading. It gives the impression that when Flcn is present, Tfe3 is active whereas Flcn physical interaction with Tfe3 actually prevents the latter to drive transcription.

Thank you for this comment, we now revised as “Tfe3, which intracellular localization and degradation is well known to be regulated by Flcn” (**lines 203-204**).

7) In lines 199-200, they wrote “The specificity of this staining was confirmed using Tfe3 deficient mice (data not shown).” Considering how important Tfe3 is for the rest of the study, I recommend the authors to include this staining control in supplementary information.

We show the Tfe3 immunostaining in *Tfe3* knockout animals (**new Supple Fig. 7a-d**). We found all the immunoreactivity including strong stromal staining seen in wild-type mice was not detected in *Tfe3* knockouts, suggesting this staining pattern is totally specific for endogenous Tfe3.

8) In line 206, they introduced their Chip-Seq experiment. It is unclear why they chose to do this experiment in the HK-2 cell line which is not endothelial. Why not use the HUVEC? Or another endothelial cell line?

As you commented we used HK-2 cells for ChIP-seq based on the technical suitability for this assay. In general, stable transfection into HUVECs is rather difficult, and it is hard to obtain large amount of cell lysates required for CHIP-seq from those cells.

Therefore, to compensate this issue, we indeed confirmed the binding of TFE3 to Prox1-0.15 in CHIP analysis (**Fig. 6a-d**).

9) In figure 6, they performed a number of experiments with siRNAs. However, they did not show anywhere that the Flcn siRNA was effectively reducing the expression of the corresponding protein.

Thank you for this comment. We now provide the immunoblotting of FLCN on HUVECs treated with si-Cont or si-*FLCN* (**new Fig. 6e**).

10) The discussion is a bit short. Although this study is focused on the vascular system, it would be interesting to compare the phenotype they observed in endothelial cells with the role of Flcn in other systems such as ESC (Betschinger et al. 2013, PMID: 23582324) or hematopoietic stem cells (Baba et al. 2016, PMID: 27095138). That would help the reader to have a broader view on the function of Flcn in different cellular contexts.

Thank you for this recommendation. Now we discussed more about the role of Flcn in other systems and relevance to our data in **lines 282-287**. We also discussed about the meaning of Prox1 expressing (LEC-biased) VECs in the formation of LV-BV misconnection as (**Lines 272-281**).

Comments of Referee #4 (blue) and our responses (black):

The authors studied a molecular mechanism that appears to maintain a separation of blood and lymphatic vessels. They studied in a mouse model the genetic deficiency of folliculin, a tumor suppressor, leading to misconnection of blood and lymphatic vessels in mice and humans. Absence of folliculin resulted in the appearance of lymphatic-biased venous endothelial cells caused by ectopic expression of Prox 1.

Interestingly, nuclear translocation of the basic helix-loop-helix (bHLH) transcription factor Transcription Factor E3 (TFE3), binding to an unprecedented regulatory element of Prox1, enhanced its venous expression. The authors conclude that folliculin acts as a gatekeeper maintaining the separation of blood and lymphatic vessels by limiting the plasticity of committed endothelial cells. Clearly, the presented data are novel and convincing and will be of great interest to others in the community and the wider field. In addition, the statistical analysis is appropriate and the level of details of the methods provided is sufficient to allow a researcher to reproduce the work. All in all, I suggest to accept the manuscript in its present form.

Thank you very much for summarizing our findings and generous evaluation. Now, we revised the paper according to other reviewers' comments and I believe the present version is even better than the former one. Thanks!

REVIEWER COMMENTS

Reviewer #2 (Remarks to the Author):

While the authors have addressed many of the issues raised, the following points definitely need to be addressed:

1. The differential expression of FLCN and TFE3 between aortic, venous and lymphatic (!) endothelial cells is essential information for this manuscript. It is acceptable if the authors derive this information from their sc-RNA seq in shown in Figure S3, but this must be shown in a (Supplementary) Figure or at least explained in the text.
2. The lack of lymphatic valves after endothelial Prox1 knockout should also be mentioned in the text. A valve defect is likely to cause problems in lymphatic transport, which could well have indirect effects on the venous phenotype observed by the authors. (Besides, Prox1 is indeed more strongly expressed in valves, but is not "almost exclusively" expressed in valves).
3. The CLDN11 staining in Fig S4b does not seem to be junctional as one might expect. Please provide a better image if possible.
4. New Supplementary Figure 1 legend b-e: Prox1idEC should be FlcnidEC?

Reviewer #3 (Remarks to the Author):

My comments have been thoroughly addressed.

Reviewer #5 (Remarks to the Author):

I feel the authors adequately addressed the concerns of Reviewer number 1.

Comments of Referee #2 (blue) and our responses (black):

While the authors have addressed many of the issues raised, the following points definitely need to be addressed.

Thank you very much for your time to evaluate our manuscript again and checking thoroughly the text. We believe your recommendation greatly improve and finalize our manuscript. Our point-by-point responses are listed below:

1. The differential expression of FLCN and TFE3 between aortic, venous and lymphatic (!) endothelial cells is essential information for this manuscript. It is acceptable if the authors derive this information from their sc-RNA seq in shown in Figure S3, but this must be shown in a (Supplementary) Figure or at least explained in the text.

According to your comment, we reanalyzed the data in scRNA-seq using Seurat. As has been shown in the previous point-by-point response, the expressions of *Flcn* and *Tfe3* were significantly higher in VECs than AECs. LECs had more abundant expression of *Flcn* than AECs, but the expression of *Tfe3* was comparable between them (**new Supplementary Figure 3f**), suggesting non-transcriptional regulation of Tfe3 like nuclear translocation and lysosomal degradation. This important point is now stated in **lines 220-226**.

2. The lack of lymphatic valves after endothelial *Prox1* knockout should also be mentioned in the text. A valve defect is likely to cause problems in lymphatic transport, which could well have indirect effects on the venous phenotype observed by the authors. (Besides, *Prox1* is indeed more strongly expressed in valves, but is not “almost exclusively” expressed in valves).

Thank you for this recommendation. We stated this phenotype in **lines 198-201**. However, we would like to mention that the phenotype of no lymphatic valves was seen in *Cdh5-Cre^{ERT2+}Prox1^{fl/fl}* and *Cdh5-Cre^{ERT2+}Flcn^{fl/fl}Prox1^{fl/fl}*, but not in *Flcnⁱ EC*. It may be easier if you look at the mesenteric lymphatics in **Fig. 2e**. Therefore the venous phenotype in *Flcnⁱ EC* is unlikely to be related to lymphatic valves.

3. The CLDN11 staining in Fig S4b does not seem to be junctional as one might expect. Please provide a better image if possible.

Thank you for this suggestion. Probably the intensity of images shown in the previous version was too low making it difficult to interpret the difference. Therefore we enhanced the contrast equally for control and *Flcnⁱ EC* for better understanding (**new Fig. S4b**).

4. New Supplementary Figure 1 legend b-e: Prox1idEC should be FlcnidEC?

We corrected this typo. Thanks!

REVIEWERS' COMMENTS

Reviewer #2 (Remarks to the Author):

The authors have sufficiently addressed the remaining concerns.